# Cell-type-specific resolution epigenetics without the need for cell sorting or single-cell biology

Elior Rahmani [1], Regev Schweiger[2,3], Brooke Rhead[4], Lindsey A. Criswell[5], Lisa F. Barcellos[6], Eleazar Eskin[1,7,8], Saharon Rosset[9], Sriram Sankararaman[1,7,8] & Eran Halperin [1,7,8,10]

High costs and technical limitations of cell sorting and single-cell techniques currently restrict the collection of large-scale, cell-type-specific DNA methylation data. This, in turn, impedes our ability to tackle key biological questions that pertain to variation within a population, such as identification of disease-associated genes at a cell-type-specific resolution. Here, we show mathematically and empirically that cell-type-specific methylation levels of an individual can be learned from its tissue-level bulk data, conceptually emulating the case where the individual has been profiled with a single-cell resolution and then signals were aggregated in each cell population separately. Provided with this unprecedented way to perform powerful large-scale epigenetic studies with cell-type-specific resolution, we revisit previous studies with tissue-level bulk methylation and reveal novel associations with leukocyte composition in blood and with rheumatoid arthritis. For the latter, we further show consistency with validation data collected from sorted leukocyte sub-types.

[1] Department of Computer Science, University of California, Los Angeles, Los Angeles, CA 90095, USA. [2] Blavatnik School of Computer Science, Tel Aviv University, Tel Aviv 6997801, Israel. [3] MyHeritage Ltd., Or Yehuda 6037606, Israel. [4] Computational Biology Graduate Group, University of California, Berkeley, Berkeley, CA 94720, USA. [5] Russell/Engleman Rheumatology Research Center, Department of Medicine, University of California, San Francisco, San Francisco, CA 94143, USA. [6] School of Public Health, University of California, Berkeley, Berkeley, CA 94720, USA. [7] Department of Human Genetics, University of California, Los Angeles, Los Angeles, CA 90095, USA. [8] Department of Computational Medicine, University of California, Los Angeles, Los Angeles, CA 90095, USA. [9] Department of Statistics, Tel Aviv University, Tel Aviv 6997801, Israel. [10] Department of Anesthesiology and Perioperative Medicine, University of California, Los Angeles, Los Angeles, CA 90095, USA. Correspondence and requests for materials should be addressed to E.R. (email: elior.rahmani@gmail.com) or to E.H. (email: ehalperin@cs.ucla.edu)

Each cell type in the body of an organism performs a unique repertoire of required functions. Hence, disruption of cellular processes in particular cell types may lead to phenotypic alterations or development of disease. This presumption in conjunction with the complexity of tissue-level ("bulk") data has led to many cell-type-specific genomic studies, in which genomic features, such as gene expression levels, are assayed from isolated cell types in a group of individuals and studied in the context of a phenotype or condition of interest (e.g., refs. [1–4]).

In fact, in order to reveal cellular mechanisms affecting disease it is critical to study cell-type-specific effects. For example, it has been shown that cell-type-specific effects can contribute to our understanding of the principles of regulatory variation[5] and the underlying transcriptional landscape of heterogeneous tissues such as the human brain[6], it can provide a finer characterization of tumor heterogeneity[7,8], and it may reveal disease-related pathways and mechanisms of genes that were detected in genetic association studies[9,10]. Moreover, these findings are typically not revealed when a heterogeneous tissue is studied. For example, in ref. [9] it has been shown that the FTO allele associated with obesity represses mitochondrial thermogenesis in adipocyte precursor cells. Particularly, in that study it is shown that the developmental regulators IRX3 and IRX5 had genotype-associated expression in primary preadipocytes, while genotype-associated expression was not observed in whole-adipose tissue, indicating that the effect was cell-type specific and restricted to preadipocytes.

In spite of the clear motivation to conduct studies with a cell-type-specific resolution, while developments in genomic profiling technologies have led to the availability of many large bulk data sets with hundreds or thousands of individuals (e.g., refs. [11–13]), cell-type-specific data sets with a large number of individuals are still relatively scarce. Particularly, cell-type-specific studies are typically drastically restricted in their sample sizes owing to high costs and technical limitations imposed by both cell sorting and single-cell approaches. This restriction is especially profound for epigenetic studies with single-cell DNA methylation—while pioneering works on single-cell methylation have demonstrated significant advances (e.g., refs. [14–17]), profiling methylation with single-cell resolution is still limited in coverage and throughput and currently cannot be practically used to routinely obtain large-scale data for population studies (the most eminent recent studies included data from only a few individuals). This, in turn, substantially limits our ability to tackle questions such as identification of disease-related altered regulation of genes in specific cell types and mapping of diseases to specific manifesting cell types.

Technologies for profiling single-cell methylation are currently still under development, and some of these attempts will potentially allow sometime in the future for the analysis of cell-type-specific methylation across or within populations. However, even if such technologies emerge in the near future, the large number of existing bulk methylation samples that have been collected by now are still an extremely valuable resource for genomic research (e.g., more than 100,000 bulk profiles to date in the Gene Expression Omnibus (GEO) alone[18]). These data reflect years of substantial community-wide effort of data collection from multiple organisms, tissues, and under different conditions, and it is therefore of great importance to develop new statistical approaches that can provide cell-type-specific insights from bulk data.

Here, we introduce Tensor Composition Analysis (TCA), a novel computational approach for learning cell-type-specific DNA methylation signals (a tensor of samples by methylation sites by cell-types) from a typical two-dimensional bulk data (samples by methylation sites). Conceptually, TCA emulates the scenario in which each individual in the bulk data has been profiled with a single-cell resolution and then signals were aggregated in each cell population of the individual separately.

We demonstrate the utility of TCA by applying it to data from previously published epigenome-wide association studies (EWAS). Particularly, we apply TCA to a previous large methylation study with rheumatoid arthritis (RA), in which DNA methylation profiles (CpG sites) were collected from cases and controls and tested for association with RA status[19]. Our analysis reveals novel cell-type-specific associations of methylation with RA without the need to collect cost prohibitive cell-type-specific data for a large number of individuals. Finally, we use independent data sets of cell-sorted methylation data to test the replicability of our results.

## Results

**Enhancing epigenetic studies with cell-type-specific resolution.** Different cell types are known to differ in their methylation patterns. Therefore, a bulk methylation sample collected from a heterogeneous tissue represents a combination of different signals coming from the different cell types in the tissue. Since cell-type composition varies across individuals, testing for correlation between bulk methylation levels and a phenotype of interest may lead to spurious associations in case the phenotype is correlated with the cell-type composition[20]. A widely acceptable solution to this problem is to incorporate the cell-type composition information into the analysis of the phenotype by introducing it as covariates in a regression analysis. Even though this procedure is useful for eliminating spurious findings, it does not take into account the fact that individuals are expected to vary in their methylation levels within each cell type (i.e., not just in their cell-type composition). Effectively, taking this approach results in an analysis that is conceptually similar to a study in which the cases and controls are matched on cell-type distribution, however, cell-type-specific signals are not explicitly modeled and leveraged.

In order to illustrate the above, consider the simple scenario, where the samples in the study are matched on cell-type distribution. Given no statistical relation between the phenotype and the cell-type composition, association studies typically assume a model with the following structure:

$$y_i = x_i\beta + \epsilon_i \quad (1)$$

Here, $y_i$ represents the phenotypic level of individual $i$, $x_i$, and $\beta$ represent the bulk methylation level of individual $i$ at a particular site under test and its corresponding effect size, and $\epsilon_i$ represents noise. This standard formulation assumes that a single parameter ($\beta$) describes the statistical relation between the phenotype and the bulk methylation level. We argue that this formulation is a major oversimplification of the underlying biology. In general, different cell types may have different statistical relations with the phenotype. Thus, a more realistic formulation would be:

$$y_i = \sum_{h=1}^{k} x_{ih}\beta_h + \epsilon_i \quad (2)$$

Here, $x_{i1}, \ldots, x_{ik}$ are the methylation levels of individual $i$ in each of the $k$ cell types composing the studied tissue and $\beta_1, \ldots, \beta_k$ are their corresponding cell-type-specific effects.

Applying a standard analysis as in Eq. (1) to bulk data may fail to detect even strong cell-type-specific associations with a phenotype. For instance, consider the scenario of a case/control study, where the methylation of one particular cell type is associated with the disease. In this scenario, due to the signals arising from other cell types, the observed bulk levels may obscure the real association and not demonstrate a difference between the cases and controls; importantly, in general, merely taking into account the variation in cell-type composition between individuals does not allow the detection of the association (Fig. 1). Thus, allowing analysis with a cell-type-specific resolution (i.e.,

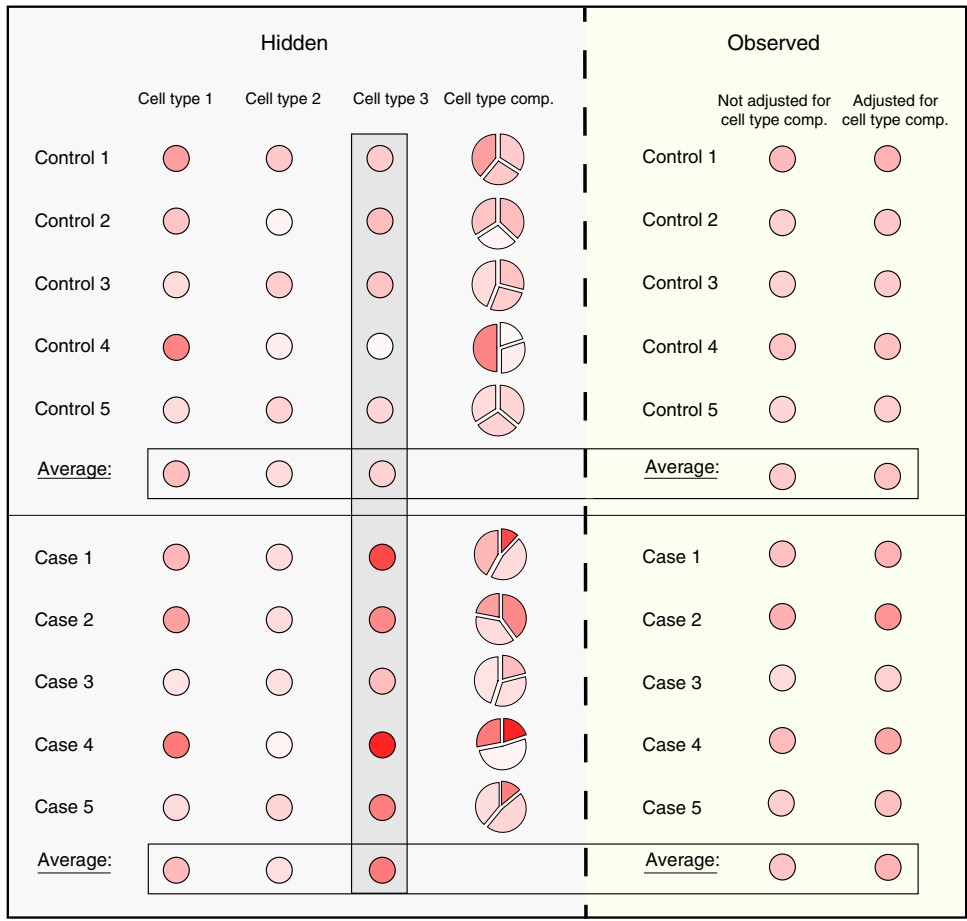

**Fig. 1** Observed bulk methylation levels may obscure cell-type-specific signals. Neither the observed methylation levels nor the observed levels after adjusting for the variability in cell-type composition can demonstrate a clear difference between cases and controls, in spite of a clear (unobserved) difference in cell type 3. Methylation levels are represented by a gradient of red color, and adjusted observed levels were calculated for each sample by removing the cell-type-specific mean levels, weighted by its cell-type composition

obtaining $x_{i1}, \ldots, x_{ik}$ for each individual $i$)—beyond being required for revealing disease-manifesting cell types—is also important for the detection of true signals.

Notably, in the context of differential gene expression analysis, it has been previously suggested that cell-type-specific effects can be estimated by treating a phenotype of interest as a covariate (i.e., of the expression level) with potentially different effects on different cell types[21,22]. Practically, this approach suggests to evaluate the effect of an interaction term (i.e., a multiplicative term) of the cell-type composition and the phenotype under a standard regression framework (i.e., by adding the interaction term to Eq. (1))[22]; equivalently, one may achieve the same goal by solving multiple decomposition problems (one for each possible value of the phenotype)[21]. In fact, this concept was recently applied and reported in the context of DNA methylation in an attempt to detect cell-type-specific differences in methylation[23]. However, as we demonstrate below, a more detailed model of the variation in bulk methylation data, as described in this manuscript, allows a substantial improvement in power.

We propose a new model for DNA methylation, where we assume that the cell-type-specific methylation levels of an individual are coming from a distribution that—up to methylation altering factors such as age[24] and sex[25]—is shared across individuals in the population. Based on this model, we developed TCA, a method for learning the unique cell-type-specific methylomes of each individual sample from its bulk data. We highlight the conceptual difference between TCA and a

traditional decomposition approach in Fig. 2, and we provide a more detailed illustration of the model in Supplementary Fig. 1. Here, we focus on the application of TCA for association studies, where we only implicitly consider the cell-type-specific methylomes of each individual by integrating over their distributions (see "Methods").

Importantly, TCA requires knowledge of the cell-type proportions of the individuals in the data. These can be computationally estimated using either a reference-based supervised approach[26] or a reference-free semi-supervised approach[27]; current reference-free unsupervised methods, however, are unable to provide reasonable estimates of cell-type proportions but rather only linear combinations of them[27]. Notably, in cases where only noisy estimates of the cell-type proportions are available (i.e., owing to inaccuracies of the computational method used for estimation), they can be used for initializing the optimization procedure of the TCA model, which can then provide improved estimates (Supplementary Fig. 2). As a result, as we show next, TCA performs well even in cases where only noisy estimates of the cell-type proportions are available.

**Detecting cell-type-specific associations using TCA.** In order to empirically verify that TCA can learn cell-type-specific methylation levels, we first leveraged whole-blood methylation data collected from sorted leukocytes[28] to simulate heterogeneous bulk methylation data. While the bulk data captured the

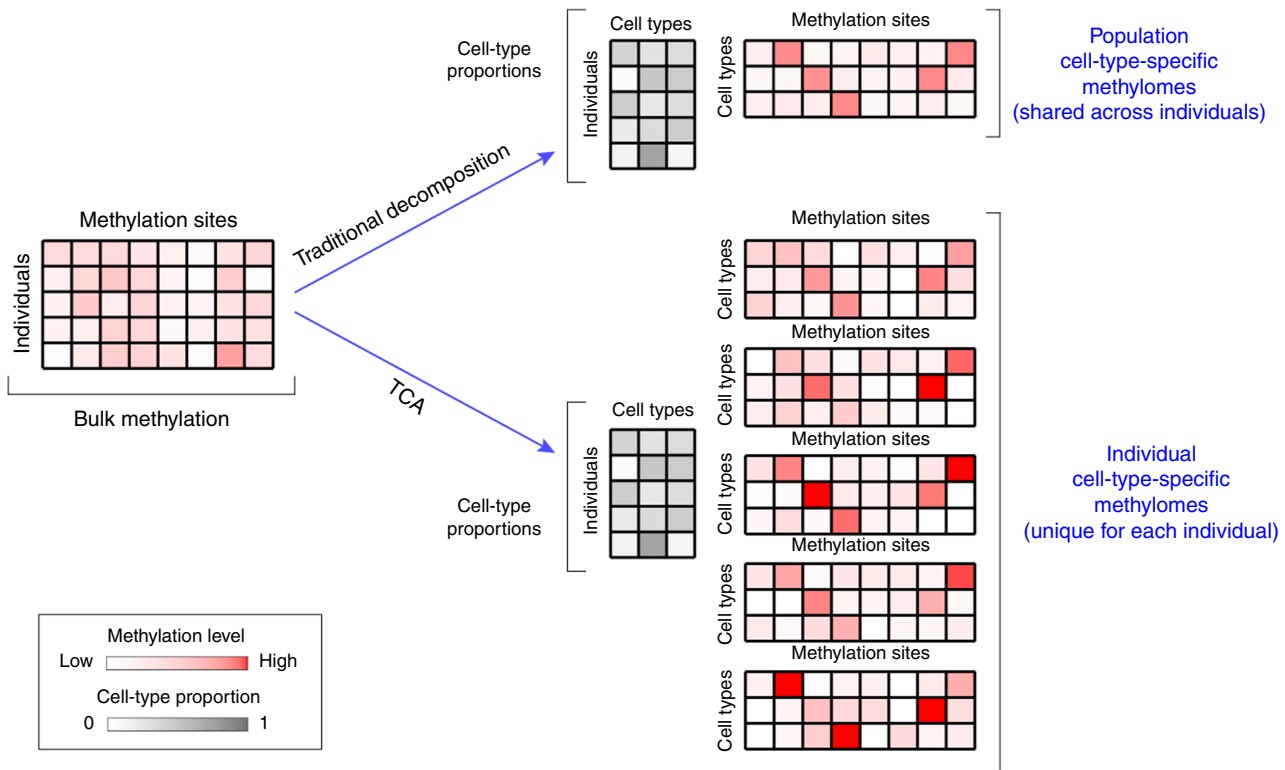

**Fig. 2** TCA versus a traditional decomposition approach. Given bulk DNA methylation data from a heterogeneous tissue, previous decomposition methods (e.g., PCA, ReFACTor[33], or a reference-based decomposition[26]) aim at estimating a matrix of the cell-type proportions of the individuals and a matrix of the cell-type-specific methylomes in the sample (shared across individuals). In contrast, TCA aims at estimating a matrix of the cell-type proportions of the individuals and—for each individual—a matrix of the unique cell-type-specific methylomes of the individual

cell-type-specific signals to some extent, as expected, TCA performed substantially better (Supplementary Figs. 3 and 4). We further observed that TCA effectively captures the effects of methylation altering covariates (Supplementary Figs. 5 and 6).

We next evaluated the performance of TCA in detecting cell-type-specific associations by simulating whole-blood methylation and corresponding phenotypes with cell-type-specific effects. We compared the performance of TCA with a standard regression analysis of the bulk levels and with the method CellDMC, an interaction-based test that was recently evaluated in the context of detecting cell-type-specific associations with methylation[23]. Notably, we provided CellDMC with the true underlying cell-type proportions as an input. Beyond introducing interaction terms into a standard regression framework, CellDMC also considers additive effects of the cell-type composition. Given the true cell-type proportions, it therefore achieves a perfect linear correction for cell-type composition. Hence, CellDMC practically reflects in our experiments an upper bound for the performance of any standard method that merely accounts for linear differences in cell-type composition across individuals.

Our experiments verify that TCA yields a substantial increase in power over the alternatives under different scenarios Particularly, in its worst performing scenario, TCA achieved a median of 2.25-fold increase in power (across all tested effect sizes) over the standard regression approach and a median of 11.15-fold increase in power in the best performing scenario (Fig. 3); compared with CellDMC, TCA achieved a median of between 1.46- and 12.25-fold increase in power across all scenarios. Repeating these experiments while including cell-type-specific affecting covariates and under a nonparametric

distribution of the cell-type proportions (i.e., rather than a parametric one) demonstrated similar results (Supplementary Fig. 7).

Remarkably, TCA demonstrated the highest improvement in a scenario where all cell types had the exact same effect size, although this is intuitively a favorable scenario for a standard regression analysis, which does not model cell-type-specific signals (Fig. 3). Interestingly, in spite of the high power achieved by TCA, we found it to be conservative (i.e., less false positives than expected; Supplementary Fig. 8); this can be explained by the optimization procedure of the model (Supplementary Methods).

Finally, we performed an additional power analysis stratified by cell types, which, once again, showed that TCA robustly outperforms the alternative approaches (Supplementary Figs. 9 and 10). This analysis further revealed that under the scenario of a single causal cell type, TCA achieved better power when the causal cell type was highly abundant (as opposed to lowly abundant); these results are expected, given that bulk signals are mostly dominated by abundant cell types. For instance, considering a moderate effect size corresponding to a signal-to-noise ratio of 1, we found that TCA achieved a median power of 1 and 0.52 in granulocytes and CD4+ cells (the two most abundant cell types; mean abundance of 0.67 and 0.11, respectively), yet only a limited power in the less abundant cell types; for example, in the two least abundant cell types considered, B cells and NK cells (mean abundance 0.03 for both), TCA could only achieve a median power of 0.08 and 0.03 under the same effect size (Supplementary Fig. 9).

**Cell-type-specific differential methylation in immune activity.** In general, the methylation levels in a particular cell type are not

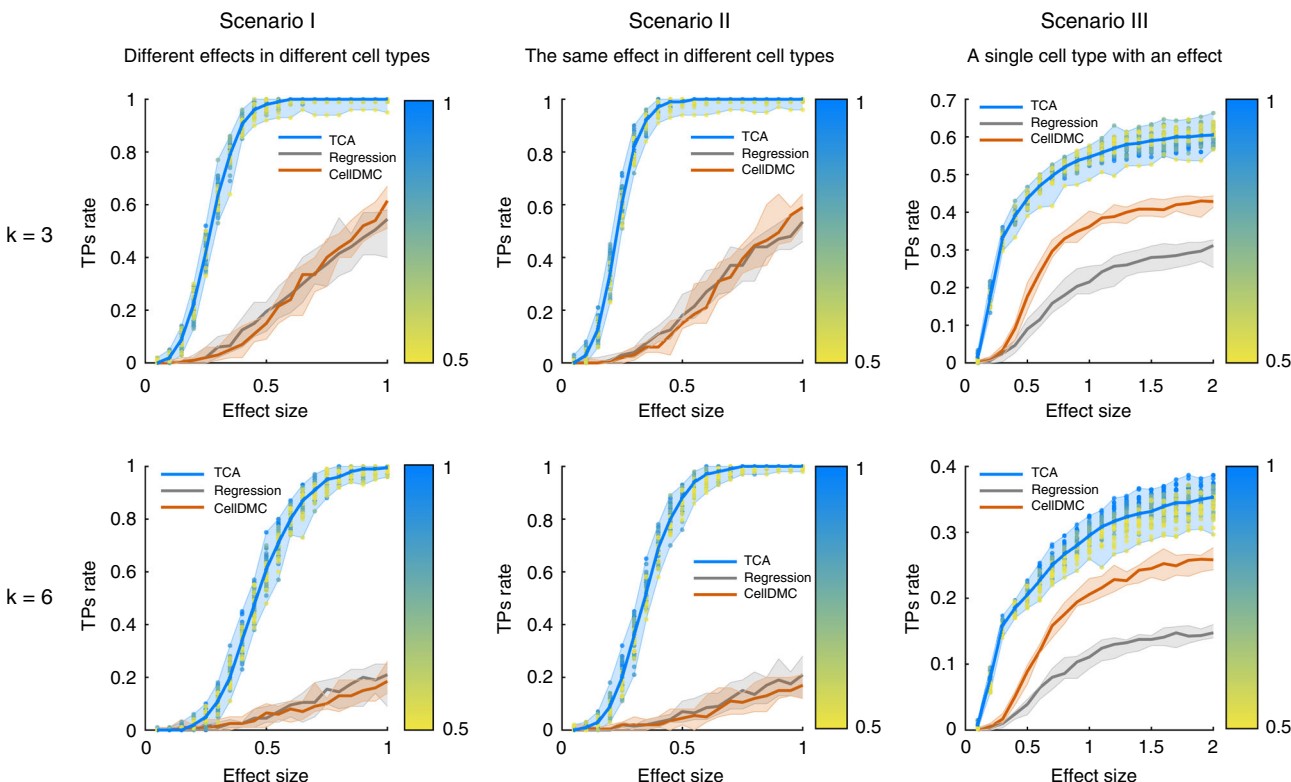

**Fig. 3** An evaluation of power for detecting cell-type-specific associations with DNA methylation. Performance was evaluated using three approaches: TCA, a standard linear regression with the observed bulk data, and CellDMC with the true cell-type proportions as an input. The numbers of true positives (TPs) were measured under three scenarios using a range of effect sizes: different effect sizes for different cell types (Scenario I), the same effect size for all cell types (Scenario II), and a single effect size for a single cell type (Scenario III); each of the scenarios was evaluated under the assumption of three constituting cell types ($k = 3$; top row) and six constituting cell types ($k = 6$; bottom row). Lines represent the median performance across 10 simulations and the colored areas reflect the results range across the multiple executions. The colored dots reflect the results of TCA under different initializations of the cell-type proportion estimates (i.e., different levels of noise injected into TCA), where the color gradients represent the mean absolute correlation of the initial estimates with the true values (across all cell types)

expected to be related to the tissue cell-type composition. Therefore, in the analysis of sorted-cell or single-cell methylation, there is no need to account for cell-type composition. In contrast, it is now widely acknowledged that in analysis of bulk methylation one has to account for cell-type composition in cases where it is correlated with the phenotype of interest[20]. For a phenotype that is highly correlated with the cell-type composition, such a correction of bulk methylation data is expected to reduce true underlying signals, potentially resulting in no findings (i.e., false negatives). As opposed to analysis of bulk data, cell-type specific analysis would not reduce the signal in this case. To demonstrate this, we consider an extreme case where the phenotype is the cell-type composition. Specifically, we defined the level of immune activity of an individual as its total lymphocyte proportion in whole-blood, and aimed at finding methylation sites that are associated with the regulation of immune activity.

Since bulk methylation data is a composition of signals that depend on to the cell-type proportions, a standard regression approach with whole-blood methylation is expected to fail to distinguish between false and true associations with immune activity. We verified this using whole-blood methylation data from a previous study by Liu et al. ($n = 658$)[19] (Supplementary Fig. 11). Importantly, accounting for the cell-type composition in this case would eliminate any true signal in the data, as the immune response phenotype is perfectly defined by the cell-type composition.

We next performed cell-type-specific analysis. Applying CellDMC resulted in a massive inflation in test statistic, which

failed to distinguish between false and true associations (Fig. 4a). Using TCA, in contrast, resulted in 8 experiment-wide significant associations ($p$-value $< 9.87e-07$; Fig. 4b and Supplementary Data 1). Importantly, 6 of the associated CpGs reside in 5 genes that were either linked in GWAS to leukocyte composition in blood or that are known to play a direct role in the regulation of leukocytes: *CD247*, *CLEC2D*, *PDCD1*, *PTPRCAP*, and *DOK2* (Supplementary Data 1). The remaining associated CpGs reside in the genes *SDF4* and *SEMA6B*, which were not previously reported as related to leukocyte composition. Using a second large whole-blood methylation data set ($n = 650$)[29], we could replicate the associations with 4 out of the 7 genes (*PTPRCAP*, *DOK2*, *SDF4*, and *SEMA6B*; $p$-value $< 0.0063$; Supplementary Data 1). Our results are therefore consistent with the possibility that methylation modifications in these genes are involved in the regulation of immune activity.

**Cell-type-specific differential methylation in rheumatoid arthritis.** RA is an autoimmune chronic inflammatory disease which has been previously related to changes in DNA methylation[30,31]. In order to further demonstrate the utility of TCA, we revisited the largest previous whole-blood methylation study with RA by Liu et al. ($n = 658$)[19].

As a first attempt to detect associations between methylation and RA status, we applied a standard regression analysis, which yielded 6 experiment-wide significant associations ($p$-value $< 2.33e-7$; Fig. 4c and Supplementary Data 2), overall in line with

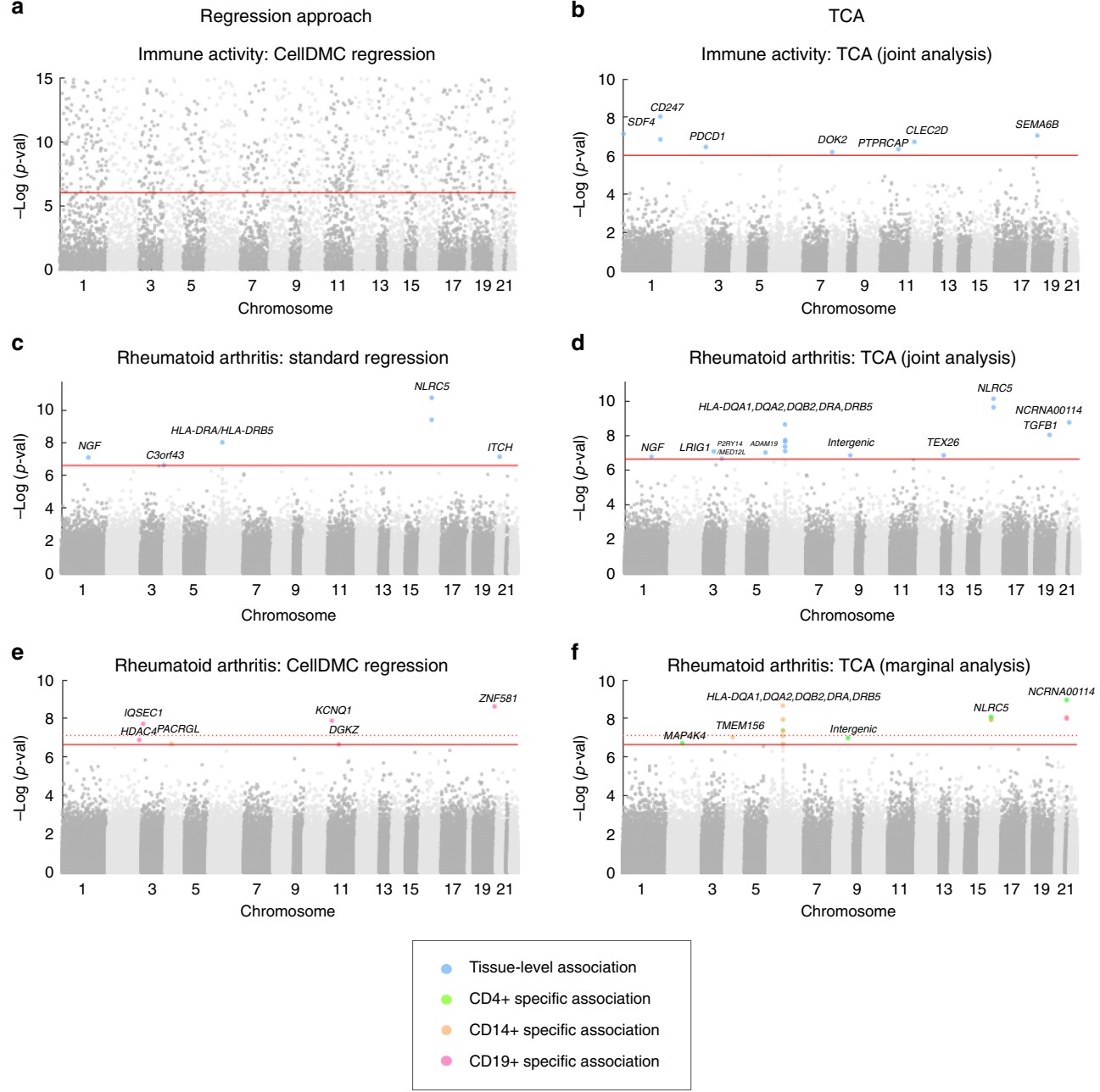

**Fig. 4** Results of the association analysis with level of immune activity and with rheumatoid arthritis in the Liu et al. whole blood methylation data, presented by Manhattan plots of the −log10 *P*-values for the association tests. **a**, **b** Shown are results with immune activity using CellDMC (results subsampled and truncated for visualization) and using TCA. **c**, **d** Shown are results of the RA analysis using standard regression and using TCA under the assumption of a single effect size for all cell types. **e**, **f** Shown are results of a cell-type-specific analysis of RA using CellDMC and using TCA. Solid horizontal red lines represent the experiment-wide significance threshold, and dotted horizontal red lines represent the significance threshold adjusted for three experiments corresponding to the three cell types

previous studies that analyzed this data set[32,33]. In order to allow an intuitive comparison with a standard regression, we performed a second analysis under the TCA model while assuming a single effect size in all cell types, which is expected to be a favorable scenario for a standard regression analysis. Remarkably, TCA found 15 experiment-wide significant CpGs, 11 of which were not reported by the standard regression analysis. Altogether, these 15 associations highlighted RA as an enriched pathway (*p*-value = 1.45e−07; Fig. 4d and Supplementary Data 2).

The presumption that only some particular immune cell-types are related to the pathogenesis of RA, have led to studies

with methylation collected from sorted populations of leukocytes (e.g., refs. [34–36]). In a recent study by Rhead et al., some of us investigated differences in methylation patterns between RA cases and controls using data collected from sorted cells[36]. Particularly, methylation levels were collected from two sub-populations of CD4+ T cells (memory cells and naive cells; *n* = 90, *n* = 88), CD14+ monocytes (*n* = 90), and CD19+ B cells (*n* = 87). Although this study involved a considerable data collection effort in an attempt to provide insights into the methylome of RA patients at a cell-type-specific resolution, it does not allow the detection of experiment-wide significant

associations (Supplementary Fig. 12), possibly owing to the limited sample size.

In order to reconcile with the sample size limitation in the sorted data by Rhead et al., we considered it for validation of the results reported by TCA in the large whole-blood data rather than for detecting novel associations. We found that 11 of the 15 CpGs reported by TCA (and 4 of the 6 CpGs reported by a standard regression) had a significant $p$-value at level 0.05 in at least one of the cell types, reflecting a high consistency with the results reported by TCA.

We next used TCA to test for associations in each of CD4+, CD14+, and CD19+ cells separately (i.e., a marginal test for each cell type, without the restriction of a single effect size). This analysis reported 15 cell-type-specific associations with 11 CpGs: 6 associations in CD4+, 8 in CD14+, and one association in CD19+ cells ($p$-value $< 2.33e{-}07$; Fig. 4f and Supplementary Data 2). Considering a more stringent significance threshold in order to account for the three separate experiments we conducted for the three cell types resulted in 10 cell-type-specific associations with 7 CpGs ($p$-value $< 7.78e{-}08$; Fig. 4f and Supplementary Data 2). We further found these CpGs to be enriched for involvement in the RA pathway ($p$-value $= 9.47e{-}07$); particularly, 4 of these CpGs reside in HLA genes (or in an intergenic HLA region) that were previously reported in GWAS as RA genetic risk loci: HLA-DRA, DRB5, DQA1, and DQA2 (Supplementary Data 2).

We further sought to evaluate the 15 associations found by the TCA marginal test using sorted data. We found that in the Rhead et al. data 4 of the 6 associations in CD4+ and 4 of the 8 associations in CD14+ had a significant $p$-value at level 0.05, with all associations having overall low $p$-values ($p$-value $\leq 0.35$ for all 15 associations; Supplementary Data 2). Following the enrichment in small $p$-values, considering a false discovery rate (FDR) criterion for the entire set of 15 associations revealed significant $q$-values at level 0.05 for all 15 associations. We further considered an additional data set with sorted CD4+ methylation from an RA study by Guo et al. ($n = 24$) and found it to be consistent ($p$-value $< 0.05$) with 3 of the 4 CD4+ associations that were verified in the Rhead et al. data.

Notably, applying CellDMC as an alternative approach for detecting cell-type-specific associations in CD4+, CD14+, and CD19+ resulted in 6 genome-wide significant hits: one in CD14+ and five in CD19+ (and only three hits in CD19+ if accounting for the three separate experiments; Fig. 4e). However, none of these 6 hits were found to be significant in the sorted cells data by Rhead et al. ($p$-value $> 0.05$), thus, echoing our conclusions from the power simulation showing a substantial gap in power between TCA and CellDMC.

Finally, we note that the lack of evidence (from the sorted cells data) for some of the associated CpGs may be explained in part by the fact that each data set was collected from a different population; specifically, Liu et al. studied a Swedish population, Rhead et al. studied a heterogeneous European population, and Guo et al. studied a Han Chinese population. In the case of TCA, another possibility is that it did not attribute the correct cell types to some of the associations. A support for this possibility is given by the fact that two associations—cg16411857 and cg22812614) were attributed to CD4+, however were supported by the sorted data to be CD14+ specific, and another association (cg11767757) was attributed to all cell types, however, was only supported by the sorted data to be CD14+ specific.

need to collect cost prohibitive cell-type-specific data. This methodology is particularly useful in light of the large number of bulk samples that have been collected by now, and due to the fact that currently single-cell methylation technologies are not practically scalable to large population studies. Importantly, we found that TCA is substantially superior to a standard regression analysis with interaction terms between the cell-type proportions and the phenotype, while adequately controlling for false positive rate, even in the case where all cell types share the same effect size. We therefore suggest that TCA should always be preferred in analysis of bulk methylation data.

Notably, a recent attempt to provide cell-type-specific context in genetic studies aims at identifying trait-relevant tissues or cell types by leveraging genetic data and known tissue or cell-type-specific functional annotations[37,38]. This approach yielded some promising results in relating trait-associated genetic loci to relevant tissues and cell types. However, it is limited to only one particular task and it is bounded by design to consider only genetic signals, whereas non-genetic signals are often also of interest in genomic studies. Moreover, this approach can only suggest an implicit cell-type-specific context by binding known annotations with heritability. In contrast, the approach taken in TCA allows the extraction of explicit cell-type-specific signals, which can potentially allow many opportunities and applications in biological research. We further note that around the time of submitting this manuscript, another model similar to TCA appeared as a preprint by Luo et al.[39] For completeness, we verified that TCA performs substantially better than the method by Luo et al. (Supplementary Figs. 13 and 14; see "Methods"); given that the latter was not published by the time of submitting this manuscript, we separate this evaluation from the main benchmarking in our work.

A potential limitation of TCA is the need for rarely available cell-type proportions as an input. We alleviate this issue by allowing TCA to get estimates of the cell-type proportions using standard methods[26,27] and then re-estimating them following the TCA model. As we showed, this allows TCA to provide good results even when just noisy estimates of the cell-type proportions are available. In practice, obtaining such estimates can be done using either a reference-based approach[26] or a semi-supervised approach[27], in case a methylation reference is not available for the studied tissue.

Our experiments and mathematical results show that TCA can extract cell-type-specific signals from abundant cell types better compared with lowly abundant cell types. Another potential limitation is expected to be in the case where the proportion of one cell type strongly covaries with the proportion of a second cell type. In case of a true association in just one of the two cell types, performing a marginal association test on each cell type separately might fail to effectively distinguish between the signals of the two cell types and report an association in both cell types. In light of these limitations, we suggest that future studies include small replication data sets from sorted or single cells. Future work might be able to alleviate this issue by modeling the covariance of the cell-type proportions.

Finally, in this paper we focus on the application of TCA to epigenetic association studies. However, TCA can be formulated as a general statistical framework for obtaining underlying three-dimensional information from two-dimensional convolved signals, a capability which can benefit various domains in biology and beyond.

## Discussion
We proposed a methodology that can reveal novel cell-type-specific associations from bulk methylation data, i.e., without the

## Methods
**Modeling cell-type-specific variation in DNA methylation**. Here, we summarize the model and mathematical methods. Further details are provided in Supplementary Methods. Since TCA can most naturally be described as a generalization

of matrix factorization, we further provide a brief technical overview of matrix factorization (Supplementary Methods).

Let $Z_{hj}^i$ denote the value coming from cell type $h \in 1, ..., k$ at methylation site $j \in 1, ... m$ in sample $i \in 1, ... n$, we assume:

$$Z_{hj}^i | \mu_{hj}, \sigma_{hj} \sim N(\mu_{hj}, \sigma_{hj}^2) \quad (3)$$

In theory, the methylation status of a given site within a particular cell is a binary condition. However, unlike in the case of genotypes, methylation status may be different between different cells (even within the same individual, site and, cell type). We therefore consider a fraction of methylation rather than a fixed binary value. In array methylation data, possibly owing to the large number of cells used to construct each individual signal, we empirically observe that a normal assumption is reasonable. Admittedly, normality may not hold for values near the boundaries (i.e., sites with mean methylation levels approaching 0 or 1); this can be addressed by applying variance stabilizing transformations such as a logit transformation (commonly referred to as $M$-values in the context of methylation)[40]. However, in practice, we ignore such consistently methylated or consistently unmethylated sites (e.g., in our experiments we discarded sites with mean value higher than 0.9 or lower than 0.1), which results in a set of sites that demonstrate an approximately linear relation with their respective $M$-values[40]. This makes the normality assumption reasonable and therefore widely accepted in the context of statistical analysis of DNA methylation.

Let $W \in \mathbb{R}^{k \times n}$ be a non-negative constant weights matrix of $k$ cell types for each of the $n$ samples (i.e., cell-type proportions; each column sums up to 1), we assume the following model for site $j$ of sample $i$ in the observed heterogeneous methylation data matrix $X$:

$$X_{ij} = \sum_{h=1}^{k} w_{hi} Z_{hj}^i + \epsilon_{ij}, \quad \epsilon_{ij} \sim N(0, \tau^2) \quad (4)$$

where $w_{hi}$ is the proportion of the $h$-th cell type of sample $i$ in $W$, and $\epsilon_{ij}$ represents an additional component of measurement noise which is independent across all samples. We therefore get that $X_{ij}$ follows a normal distribution with parameters that are unique for each individual $i$ and site $j$. Put differently, we assume that the entries of $X$ are independent but also different in their means and variances.

**Tensor Composition Analysis (TCA).** Following the assumptions in (3) and (4), the conditional probability of $Z_j^i = \left(Z_{1j}^i, ..., Z_{kj}^i\right)^T$ given $X_{ij}$ can be shown (Supplementary Methods) to satisfy

$$Pr(Z_j^i = z_j^i | X_{ij} = x_{ij}, w_i, \mu_j, \sigma_j, \tau)$$
$$\propto exp\left(-\tfrac{1}{2}(a_{ij} - z_j^i)^T S_{ij}^{-1}(a_{ij} - z_j^i)\right) \quad (5)$$

where

$$\Sigma_j = diag(\sigma_{1j}^2, ..., \sigma_{kj}^2) \quad (6)$$

$$S_{ij} = \left(\frac{w_i w_i^T}{\tau^2} + \Sigma_j^{-1}\right)^{-1} \quad (7)$$

$$a_{ij} = S_{ij}\left(\frac{x_{ij}}{\tau^2} w_i + \Sigma_j^{-1} \mu_j\right) \quad (8)$$

Essentially, our suggested method, TCA, leverages the information given by the observed values $\{x_{ij}\}$ for learning a three-dimensional tensor consisted of estimates of the underlying values $\{z_{hj}^i\}$. This is done by setting the estimator $\hat{z}_j^i$ to be the mode of the conditional distribution in (5):

$$\hat{z}_j^i = a_{ij} = \left(\frac{w_i w_i^T}{\tau^2} + \Sigma_j^{-1}\right)^{-1}\left(\frac{x_{ij}}{\tau^2} w_i + \Sigma_j^{-1} \mu_j\right) \quad (9)$$

TCA requires the cell-type proportions $W$ as an input. Given $W$, the parameters $\tau, \{\mu_j\}, \{\sigma_j\}$ can be estimated from the observed data under the assumption in (4). In practice, the cell-type proportions are typically unknown. In such cases, $W$ can be estimated computationally using standard methods (e.g., refs. [26,27]) and then re-estimated under the TCA model in an alternating optimization procedure with the rest of the parameters in the model. The TCA model can further account for covariates, which may either directly affect $Z_j^i$ (e.g., age and sex) or affect the mixture $X_{ij}$ (e.g., batch effects). For more details and a full derivation of the conditional distribution of $Z_j^i$, while accounting for covariates, and for information about parameters inference see Supplementary Methods.

In order to see why TCA can learn non-trivial information about the $\{z_{hj}^i\}$ values, consider a simplified case where $\tau = 0$, $\mu_{hj} = 0$, $\sigma_{hj} = 1$ for each $h$ and a specific given $j$. In this case, it can be shown (Supplementary Methods) that

$$Z_{hj}^i | X_{ij} = x_{ij} \sim N\left(\frac{w_{hi} x_{ij}}{\sum_{l=1}^{k} w_{li}^2}, 1 - \frac{w_{hi}^2}{\sum_{l=1}^{k} w_{li}^2}\right) \quad (10)$$

That is, given the observed value $x_{ij}$, the conditional distribution of $Z_{hj}^i$ has a lower variance compared with that of the marginal distribution of $Z_{hj}^i$ ($\sigma_{hj}^2 = 1$), thus

reducing the uncertainty and allowing us to provide non-trivial estimates of the $\{z_{hj}^i\}$ values. This result further implies that in the context of DNA methylation, where the weights matrix $W$ corresponds to a matrix of cell-type proportions, we should expect to gain better estimates for the $\{z_{hj}^i\}$ levels in more abundant cell types compared with cell types with typically lower abundance. For more details see Supplementary Methods.

**Applying TCA to epigenetic association studies.** We next consider the problem of detecting statistical associations between DNA methylation levels and biological phenotypes. Let $X \in \mathbb{R}^{n \times m}$ be an individuals by sites matrix of methylation levels, and let $Y$ denote an $n$-length vector of phenotypic levels measured from the same $n$ individuals; typical association studies usually consider the following model for testing a particular site $j$ for association with $Y$:

$$Y_i = X_{ij}\beta_j + e_i, \quad e_i \sim N(0, \sigma^2) \quad (11)$$

where $Y_i$ is the phenotypic level of individual $i$, $\beta_j$ is the effect size of the $j$-th site, and $e_i$ is a component of i.i.d. noise. For the convenience of presentation, we omit potential covariates which can be incorporated into the model. In a typical EWAS, we fit the above model for each feature, and we look for all features $j$ for which we have sufficient statistical evidence of non-zero effect size (i.e., $\beta_j \neq 0$).

In principle, one can use TCA for estimating cell-type-specific levels, and then look for cell-type-specific associations by fitting the model in (11) with the estimated cell-type-specific levels (instead of directly using $X$). However, an alternative one-step approach can be also used. This approach leverages the information we gain about $z_{hj}^i$ given that $X_{ij} = x_{ij}$ for directly modeling the phenotype as having cell-type-specific effects. Specifically, consider the following model:

$$Y_i = Z_{lj}^i \beta_{lj} + e_i, e_i \sim N(0, \phi^2) \quad (12)$$

where $\beta_{lj}$ denotes the cell-type-specific effect size of some cell type of interest $l$. Provided with the observed information $x_{ij}$, while keeping the assumptions in (3) and (4), it can be shown (Supplementary Methods) that:

$$Y_i | X_{ij} = x_{ij} \sim N\left(\beta_{lj}\left(\mu_{lj} + \frac{w_{li}\sigma_{lj}^2 \tilde{x}_{ij}}{\tau^2 + \sum_{h=1}^{k} w_{hi}^2 \sigma_{hj}^2}\right), \phi^2 + \beta_{lj}^2\left(\sigma_{lj}^2 - \frac{w_{li}^2 \sigma_{lj}^4}{\tau^2 + \sum_{h=1}^{k} w_{hi}^2 \sigma_{hj}^2}\right)\right) \quad (13)$$

$$\tilde{x}_{ij} = x_{ij} - \sum_{h=1}^{k} w_{hi}\mu_{hj} \quad (14)$$

This shows that directly modeling $Y_i | X_{ij}$ effectively integrates the information over all possible values of $Z_{lj}^i$. Given $W$, $\mu_j$, $\sigma_j$, $\tau$ (typically estimated from $X$; Supplementary Methods), we can estimate $\varphi$ and the effect size $\beta_{lj}$ using maximum likelihood. The estimate $\hat{\beta}_{lj}$ can be then tested for significance using a generalized likelihood ratio test. Similarly, we can consider a joint test for the combined effects of more than one cell type. A full derivation of the statistical test is described in Supplementary Methods. In this paper, whenever association testing was conducted, we used this direct modeling of the phenotype given the observed methylation levels.

Finally, we note that in principle one can also use the model in Eq. (4) for testing for cell-type-specific associations by treating the phenotype of interest as a covariate and estimating its cell-type-specific effect size. However, TCA provides a way to deconvolve the data into cell-type-specific levels, which is of independent interest beyond the specific application for association studies. Moreover, model directionality often matters, and the TCA framework allows us to directly model the phenotype rather than merely treat it as another covariate. Particularly, in the context of this paper, it is known that methylation levels are actively involved in many cellular processes such as regulation of gene expression[41], thus, making DNA methylation a potential contributing determinant in disease (which further justifies the modeling of the phenotype as an outcome).

**Implementation of TCA.** A Matlab implementation of TCA was used for deriving all the results in this paper, and an additional implementation in R was deposited as a CRAN package ("TCA"). The source code of both implementations is available from GitHub at http://github.com/cozygene/TCA.

TCA requires for its execution a heterogeneous DNA methylation data matrix and corresponding cell-type proportions for the samples in the data. In case where cell counts are not available, TCA can take estimates of the cell-type proportions, which are then optimized with the rest of the parameters in the model. For the real data experiments, we used GLINT[42] for generating initial estimates of the cell-type proportions for the whole-blood data sets. GLINT provides estimates according to the Houseman et al. model[26], using a panel of 300 highly informative methylation sites in blood[43] and a reference data collected from sorted blood cells[28]. Given these estimates, we used the TCA model to re-estimate the cell-type proportions using the top 500 sites selected by the feature selection procedure of ReFACTor[33].

**Data simulation**. We first estimated cell-type-specific means and standard deviations in each site using reference data of methylation levels collected from sorted blood cells[28]. Since we expected cell-type-specific associations to be mostly present in CpG sites that are highly differentially methylated across different cell types, we considered cell-type-specific means and standard deviations from sites which demonstrated the highest variability in cell-type-specific mean levels across the different cell types. Using the estimated parameters of a given site, we generated cell-type-specific DNA methylation levels using normal distributions, conditional on the range [0, 1]. In cases where covariates were simulated to have an effect on the cell-type-specific methylation levels, the means of the normal distributions were tuned for each sample to account for its covariates and the corresponding effect sizes (shared across samples; Supplementary Methods).

We generated cell-type proportions for each sample using a Dirichlet distribution with parameters set according to previous estimates from cell counts of 6 blood cell types[27]: 15.0727, 1.8439, 2.5392, 1.7934, 0.7240, and 0.7404, which correspond to Dirichlet parameters for granulocytes, monocytes, and 4 sub-types of lymphocytes (CD4+, CD8+, B and NK cells). In the case of three constituting cell types (granulocytes, monocytes, and lymphocytes), we set the Dirichlet parameter of lymphocytes to be the sum of the parameters of all the lymphocyte sub-types. For the experiments with a nonparametric distribution of the cell-type proportions we sampled proportions of individuals from a pool of reference-based estimates that were estimated using a reference-based method[26] for samples in two data sets (described below)[19,29].

Eventually, for each sample, we composed its methylation level at each site by taking a linear combination of the simulated cell-type-specific levels of that site, weighted by the cell composition of that sample, and added an additional i.i.d. normal noise conditional on the range [0, 1] to simulate technical noise ($\tau = 0.01$). In cases where covariates were simulated to have a global effect on the methylation levels (i.e., non-cell-type-specific effect, such as batch effects), we further added an additional component of variation for each sample according to its global covariates and their corresponding effect sizes.

**Data sets**. We used a total of 5 methylation data sets, all of which were collected using the Illumina 450K human DNA methylation array and are available from the Gene Omnibus Database (GEO). In more details, we used 3 methylation data sets that were previously collected in RA studies: a whole-blood data set by Liu et al. of 354 RA cases and 332 controls (GEO accession GSE42861)[19], a CD4+ methylation data set of 12 RA cases and 12 controls with matching age and sex (for each RA patient, a control sample with matching age and sex was collected) by Guo et al. (GEO accession GSE71841)[35], and cell-sorted methylation data collected from 63 female RA patients and 31 female control subjects in CD4+ memory cells, CD4+ naive cells, CD14+ monocytes, and CD19+ B cells (a total of 371 samples across four cell sub-types; GEO accession GSE131989); these cell-sorted data were originally described by Rhead et al.[36]. In addition, for replicating the association results with immune activity, we used another data set that was previously studied by Hannum et al. in the context of aging rates ($n = 656$; GEO accession GSE40279)[29]. Finally, for the simulation experiments we used methylation reference of sorted leukocyte cell types collected in 6 individuals from the (GEO accession GSE35069)[28].

We processed the data similarly to a recently suggested normalization pipeline[44]. Specifically, we processed the raw IDAT files of the Liu et al. data set[19] and the Rhead et al. data set[36] (each cell sub-type separately) using the "minfi" R package[45] as follows. We removed 65 SNP markers and applied the Illumina background correction to all intensity values, while analyzing probes coming from autosomal and non-autosomal chromosomes separately. We considered a threshold of $10e{-}16$ for the detection $p$-value of intensity values; probes with $p$-values higher than this threshold were treated as missing values, and samples with call rate <95% and probes with call rate <90% were excluded. Since IDAT files were not made available for the Hannum et al. data[29] and the Guo et al. data[35], we used the methylation intensity levels published by the authors. For each data set, we then performed a quantile normalization of the methylation intensity levels, subdivided by probe type, probe sub-type, and color channel, and imputed missing values using the "impute" R package (using the function impute.knn). Eventually, we calculated beta-normalized methylation levels based on the normalized intensity levels (according to the recommendation by Illumina).

We further excluded samples from the above data sets as follows. In the Liu et al. data set, we excluded two samples that demonstrated extreme values in their first two principal components (over four empirical standard deviations) and two more of the remaining samples that were regarded as outliers in the original study of Liu et al. In the Rhead et al. data set, we excluded a small batch that consisted of only 4 individuals, and in the Hannum et al. data set we removed six samples that demonstrated extreme values in their first two principal components (over four empirical standard deviations). The final numbers of samples remained for analysis in the Liu et al. data set, the Hannum data set and the Guo et al. data set were $n = 658$, $n = 650$, and $n = 24$, respectively. The numbers of samples remained for analysis in the Rhead et al. data were $n = 89$, $n = 88$, $n = 90$, and $n = 86$ for the CD4+ memory cells, CD4+ naive cells, monocytes, and B cells, respectively.

Finally, for the association experiments, we discarded consistently methylated probes and consistently unmethylated probes from the data (mean value higher than 0.9 or lower than 0.1, respectively, according to the Liu et. al discovery data), and we

further used GLINT[42] to exclude from the data CpGs coming from the non-autosomal chromosomes, as well as polymorphic and cross-reactive sites, as was previously suggested[46].

**Power simulations**. We simulated data and sampled for each site under test a normally distributed phenotype with additional effects of the cell-type-specific methylation levels of the site. We set the variance of each phenotype to the variance of the site under test, in order to eliminate the dependency of the power in the variance of the tested site (and therefore allow a clear quantification of the true positives rate under a given effect size). Particularly, when simulating an effect coming from a single cell type, we randomly generated a phenotype from a normal distribution with the variance set to the variance of the site under test in the specific cell type under test. Similarly, when simulating effects coming from all cell types, we randomly generated a phenotype from a normal distribution with the variance set to the total variance of the site under test (i.e., across all cell types).

We performed the power evaluation using simulated data with 3 constituting cell types ($k = 3$) and using simulated data with 6 constituting cell types ($k = 6$). We considered three scenarios across a range of effect sizes as follows: different effect sizes for different cell types (using a joint test), the same effect size for all cell types (using a joint test, under the assumption of the same effect for all cell types), and a scenario with only a single associated cell type (a marginal test). In the first scenario, effect sizes for the different cell types were drawn from a normal distribution with the particular effect size under test set to be the mean (with standard deviation $\sigma = 0.05$), and in the third scenario we evaluated the aggregated performance of all the marginal tests across all constituting cell types in the simulation. We further repeated the marginal test while stratifying the evaluation by cell type (i.e., the marginal test was performed under the third scenario for each cell type separately). In each of these experiments, we calculated the true positives rate of the associations that were reported as significant while adjusting for the number of sites in the simulated data.

For each scenario and for each number of constituting cell types, we simulated 10 data sets, each included 500 samples and 100 sites. Importantly, throughout the simulation study, we considered for each simulated data set the case where only noisy estimates of the cell-type proportions are available (and therefore need to be re-estimated together with the rest of the parameters in the TCA model). Specifically, for each sample in the data we replaced its cell-type proportions with randomly sampled proportions coming from a Dirichlet distribution with the original cell-type proportions of the individuals as the parameters. For each level of noise, these parameters were multiplied by a factor that controlled the level of similarity of the sampled proportions to the original proportions. Finally, for evaluating false positives rates, we followed the above procedure, however, without adding additional effects coming from methylation levels. We evaluated the false positives rate by considering the fraction of sites with $p$-value < 0.05.

**Analysis of immune activity**. We used the Liu et al. data[19] as the discovery data ($n = 658$) and the Hannum et al. data[29] as the replication data ($n = 650$). Since we expected to observe associations with the regulation of cell-type composition in CpGs that demonstrate differential methylation between different cell types, we considered for this analysis only CpGs that were reported as differentially methylated between different whole-blood cell types[20]. Specifically, we considered the sites in the intersection between the set of Bonferroni-significant CpGs that were reported as differentially methylated in whole-blood and the available CpGs in both the discovery and replication data sets; this resulted in a set of 50,123 CpGs that were available for this analysis.

We performed a standard linear regression analysis using GLINT[42] and a TCA analysis under the assumption of the same effect size in all cell types. In the analysis of the Liu et al. data we controlled for RA status, gender, age, smoking status, and known batch information, and in the analysis of the Hannum et al. data we controlled for gender, age, ethnicity, and the first two EPISTRUCTURE principal components[47] in order to account for the population structure in this data set. In both data sets, in order to take into account potentially unknown technical confounding effects, we further included the first 10 principal components calculated from the intensity levels of a set of 220 control probes in the Illumina methylation array, as suggested by Lehne et al.[44] in an approach similar to the remove unwanted variation method (RUV)[48]. These probes are expected to demonstrate no true biological signal and therefore allow to capture global technical variation in the data.

In the replication analysis, we applied a Bonferroni threshold in reporting significance, controlling for the number of genome-wide significant associations that were reported in the discovery data. The results are summarized in Supplementary Data 1, where additional description for the associated genes is provided from GeneCards[49], the GWAS catalog[50], and GeneHancer[51].

**Analysis of rheumatoid arthritis**. We used the Liu et al. data[19] as the discovery data ($n = 658$, 214,096 CpGs). We applied a standard logistic regression analysis with the RA status as an outcome using GLINT[42] and TCA analysis: under the assumption of a single effect for all cell types (joint test), and for each of CD4+, CD14+, and CD19+, under the assumption of a single associated cell type (marginal test). In every analysis, we accounted for the same variables described in the immune activity

analysis with this data set. In the TCA analysis, we additionally accounted for the first six ReFACTor components[33], calculated according to the most recent updated guidelines[52]. In order to test the associations reported by TCA for enrichment for the RA pathway, we used missMethyl[53], an R package that allows to run enrichment analysis for disease directly on CpGs (while accounting for gene length bias).

In the validation analysis with the Rhead et al. data, we applied a standard logistic regression analysis using GLINT[42] on each of the CD14+ ($n = 90$) and CD19+ ($n = 86$) data sets, while accounting for age, smoking status, and batch information. Since the Rhead et al. data included sorted-cell methylation from two sub-types of CD4+, for the replication analysis of CD4+ ($n = 81$) we performed for each site a logistic regression analysis using both its CD4+ naive cells methylation levels and CD4+ memory cells methylation.

Taking a standard regression approach in the analysis of the Guo et al. CD4+ sorted methylation data resulted in a severe inflation in test statistic. Since the cases and controls in the sample were matched for age and sex, we suspected that technical variation might have led to this inflation. In order to test that, we calculated the first principal component of control probes, similarly to the approach taken in the analysis of the Liu et al. data. However, since IDAT files were not available for the Guo et al. data, and therefore the same set of 220 control probes that were used in the Liu et al. data were not available, we used the methylation intensity levels of the 220 sites with the least variation in the data as control probes. Indeed, we found that the first PC of the control probes corresponds to the case/control status in the data almost perfectly ($r = 0.91$, $p$-value $= 6.29e{-}10$). As a result, $p$-values obtained using a standard analysis of the Guo et al. data set are not reliable. We therefore considered the following nonparametric procedure. We ranked the sites according to their absolute difference in mean methylation levels between cases and controls, and considered a simple enrichment test, wherein the $p$-value of a site was determined as its rank divided by the total number of sites in the ranking.

The results are summarized in Supplementary Data 2, where additional description for the associated genes is provided from GeneCards[49], the GWAS catalog[50], and GeneHancer[51].

**Application of CellDMC and HIRE**. We applied CellDMC using the corresponding R package by Zheng et al.[23], and provided it with the true cell-type proportions as an input throughout our simulation study, and with the same covariates we used for TCA in the real data analysis. We further applied HIRE using the corresponding R package by Luo et al.[39]. Unlike CellDMC, HIRE treats the cell-type proportions as parameters that are being estimated as part of the optimization process. Therefore, in order to provide it with a similar advantage to CellDMC, which was given access to the true cell-type proportions in the simulation study, we assigned the initial cell-type proportion estimates in the HIRE code to be the true cell-type proportions.

Since both CellDMC and HIRE provide only test statistics and $p$-values for the effects of individual cell types (i.e., only for marginal tests and not for a joint, CpG-level test), in the power simulations with effects in multiple cell types we considered a CpG to be associated with the phenotype if it had a significant association with at least one of the cell types. To make our benchmarking of TCA with these methods conservative, we allowed a favorable procedure for CellDMC and HIRE in these cases by not accounting for the number of cell types (i.e., just for the number of CpGs) when calculating true positive rates.

**Reporting summary**. Further information on research design is available in the Nature Research Reporting Summary linked to this article.

## Data availability

The DNA methylation data sets used and analyzed in this study are available in the Gene Expression Omnibus (GEO) repository under the following accession IDs: GSE42861[19], GSE71841[35], GSE40279[29], GSE35069[28], and GSE131989[36].

## Code availability

An R package named "TCA" is available from CRAN. The source code of both the R version and the Matlab version of TCA are available from GitHub under the GPL-3 license: https://github.com/cozygene/TCA.

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

## Acknowledgements

E.H. and E.R. were partially supported by NSF grant 1705197. E.H. was also partially supported by NIH grant 1R01MH115979. E.R. and R.S. were supported in part by the Israel Science Foundation (Grant 1425/13) and by the Edmond J. Safra Center for Bioinformatics at Tel-Aviv University. S.S. was supported in part by NIH grants R00GM111744, R35GM125055, NSF Grant III-1705121, an Alfred P. Sloan Research Fellowship, and a gift from the Okawa Foundation. B.R., L.A.C., and L.F.B. were supported by the Rheumatology Research Foundation (Within Our Reach grant and Health Professional Research preceptorship), the Arthritis Foundation, the Rosalind Russell/Ephraim P. Engleman Rheumatology Research Center, and the University of California–Stanford Arthritis Center of Excellence, which is funded in part by the Arthritis Foundation.

## Author contributions

E.R. and E.H. conceived and designed the project. E.R. performed data analysis. R.S., E.E., S.R., and S.S. contributed expertise. B.R., L.A.C. and L.F.B. generated and contributed data. E.R. and E.H. drafted the manuscript. All authors read and approved the manuscript.

## Additional information

**Competing interests:** The authors declare no competing interests.

