## [Peer Review File · Nature Communications]

Reviewers' comments:

Reviewer #1 (Remarks to the Author):

The manuscript from Rahmani et al presents a novel method that advances the resolution of genome-scale DNA methylation analyses to understand cell-type-specific alterations in a complex mixture. Overall, the authors present strong work that addresses a current gap in available approaches to attribute observed DNA methylation alterations in samples from mixed cell types (e.g. peripheral blood), to specific components of the mixtures.

- The normality assumptions introduced in the methods (line 277) for methylation data are not particularly well substantiated, DNA methylation data is often logit transformed to better approximate a normal distribution.
- The authors use Reinius data for the basis of simulation, and mention age and sex as related with DNA methylation. Reinius cell-type-specific data are from exclusively male subjects.
- There are some limitations of using the three dimensional tensor in the application setting of peripheral blood where there are approximately six major components. There is a use of $k=6$ in Figure 3, but additional detail regarding the noted limitation in the discussion (line 250) regarding "lowly abundant" cell types would add value.
- Why would the authors assume that each cell type has the same effect size if there are known differences in abundance and methylation is usually present or absent?
- It appears that the TCA approach testing cell-specific methylation does not inflate multiple comparisons. Please comment and add to strengths as appropriate
- Characterizing cell proportion estimates in multiple places as potentially noisy or "just moderately reasonable" is unclear, does it mean to suggest limited accuracy of the standard Reinius library applied using the Houseman approach?
- For the power simulations I am troubled by the replacement of each sample with randomly sampled proportions, this may result in overestimates of power as the true biological distribution would not be random within the Dirichlet.
- Similarly, the methods indicate better estimates in "more abundant" cell types (308-309), can the authors be more specific?
- The details of what the TCA modeling approach provides as outputs are not entirely clear.
- Language "Standard approach" is often used and in certain places not clear, specify where possible (e.g. line 125, 230-231)
- The application of the approach is interesting, specifying the modeling approaches used in figure 4 panels would be helpful. An application to data with
- Another noted limitation in the discussion is covariance of cell type abundance, given the Dirichlet distribution this will often be the case and prior knowledge of how specific cell types co-vary could in the future help address the limitation and could be noted.

- Additional opportunities for implementation of the method (R) would be most welcomed

Reviewer #2 (Remarks to the Author):

In this manuscript, Rahmani et al. proposed an epigenome-wide association study in the cell-type-specific resolution using a so-called Tensor Composition Analysis (TCA) approach. The major contribution of their work is the using of bulk data instead of cell sorting or single-cell data, which are currently infeasible in large-scale study. The method they used, TCA, is a natural generalization (3D version) of the typical Nonnegative Matrix Factorization, which is widely explored in many epigenetic research fields (Houseman et al., 2012, 2016; Lutsik et al, 2017; Onuchic et al., 2016; Rahmani et al, 2018). I have the following major concerns, mainly in the methodology part.

1. The authors made a normality assumption on the methylation level of a given CpG sites within a particular type of cells. It is not exactly true to the best of my knowledge. Methylation levels vary between 0 and 1, and thus are more suitable to be modeled as the Beta distribution. If assuming the normal distribution, the authors should do some sort of data transformation on the raw beta values first, e.g., logit or arcsine.
2. TCA require the cell-type proportion information as the input. However, in most cases, this kind of information is not available. There are two types of methods for estimation cell-type proportions from DNA methylation data. One is called 'reference-based', which is accurate but needs reference profiles of the constitutional components, and thus only available for blood samples or infiltrating immune cells. The other does not require the reference data but the estimation accuracy is quite low. So under such circumstance, the method proposed in this manuscript only works for blood samples.
3. Along the second concern, if the estimation of cell-type proportion is not reliable, how about the accuracy of the EWAS result? The authors should show the robustness of their method under different sources and levels of noise.
4. In the real data application part, how did the authors determine the optimal value of k , i.e., the number of components in tissue mixture?
5. The authors should specify the computational complexity of their method and compare with other EWAS methods.

Minor point:

1. Page 8: Index of Supplementary figure S4 is missing.
2. Page 9: Figure 2 is vague. A carefully designed and more informative workflow of the method is needed.

Reviewer #3 (Remarks to the Author):

The manuscript entitled "Cell-type-specific resolution epigenetics without the need for cell sorting or single-cell biology" by Elijah Rahmani and colleagues proposed Tensor Composition Analysis (TCA), a new mathematical approach, to identify cell-type specific methylation changes from bulk epigenetic data without cell sorting. Cellular heterogeneity is a major problem impeding the application and interpretation of epigenome-wide association studies (EWAS). It would be of great importance if cell-type specific signatures could be extracted and explored from bulk data. While the topic is very important and the idea is interesting, there are several major issues with the manuscript.

Major issues:

1. Benchmarking and a more detailed data analysis are needed. For instance, the authors indicated poor performance of TCA for lowly abundant cell types and for the situation of strongly correlated cell

types in the discussion. However, detailed evaluation on these situations using simulated data were not provided.

2. In the past several years, numerous mathematical approaches have been proposed for cell type deconvolution and subsequent ewas analyses, including both reference-based (eg. CIBERSOFT, RPC) and reference-free (eg. SVA, RefFreeEWAS, ISVA, EWASher, ReFACTor, RefFreeCellMix) approaches. They all performed better compared to a simple linear regression model. Even though these approaches assume the same effect size for all cell types, it is critical for the authors to evaluate the performance of TCA in the context of these for Scenario II in Figure 3.

3. It is unclear to me how the TCA works based on Figure 2 and its legend. Even though it is described in details in the supplementary note, it is important for readers to understand the main idea of TCA from the main text and the figure.

4. The author claimed in the abstract that “we show mathematically and experimentally that cell-type-specific methylation levels of an individual can be learned from its tissue-level bulk data, as if the sample has been profiled with a single-cell resolution and then signals were aggregated in each cell population separately”. This is a gross overstatement. First, no experiments were performed in the manuscript. Second, the results generated from TCA model were not compared to results from single cell data. In fact, the TCA model makes the normality assumption for methylation, while methylation in a single cell follows binary distribution. Thus, the TCA model, theoretically, could not be extended to single cell.

5. In the reference-free situation, it is unclear to me how TCA extracted cell-type specific methylation signature was assigned to one particular cell type when there is another cell type with a similar cell proportion. This is particularly worrisome when looking at cell-type-specific analysis using TCA in Figure 4f. Several significant CpGs identified from this approach, but failed to be replicated in the claimed cell type using sorted data, were actually significant in other cell types (eg. cg11767757 in CD14+ and CD19+ cells; cg16411857 in CD4+ cells). To me, this cannot be explained by the small sample size as claimed by the authors.

6. The most unique feature of TCA, as claimed by the authors, is the ability to identify cell-type specific methylation changes. However, based on the results from Figure 4d and 4f, cell-type specific approach did not identify any new significant hits outside of the HLA region. Combining with the fact of high false positive rate of assigned cell types by TCA as discussed in point 5, this advantage of TCA model is debatable.

7. The analysis in the last paragraph of the main text (page 16), analyzing the associations between methylation and expression, is quite problematic. The three CpG sites investigated are all located in the HLA region. There are many SNPs in the probes for these CpGs, which will greatly affect the methylation measurements. Similarly, the expression data used for analysis was generated from microarrays and the high genetic polymorphism within the HLA region will also impact the evaluation of gene expression. Thus, genotype, as a confounder, will lead to observed associations between methylation and gene expression.

Dear Editors and Reviewers,

We thank the reviewers for their comments. We have revised the manuscript to address the reviewers' questions and suggestions. Below we provide a point-by-point response to each of the reviewers' comments (our responses are in regular fonts). In addition, we include a manuscript file where the changes in the text are highlighted in a blue font.

Reviewer #1 (Remarks to the Author):

The manuscript from Rahmani et al presents a novel method that advances the resolution of genome-scale DNA methylation analyses to understand cell-type-specific alterations in a complex mixture. Overall, the authors present strong work that addresses a current gap in available approaches to attribute observed DNA methylation alterations in samples from mixed cell types (e.g. peripheral blood), to specific components of the mixtures.

- The normality assumptions introduced in the methods (line 277) for methylation data are not particularly well substantiated, DNA methylation data is often logit transformed to better approximate a normal distribution.

We thank the reviewer for pointing out this important issue. We agree that applying a variance stabilizing transformation to methylation levels (commonly referred to as beta values in the context of methylation) will generally provide a better approximation of a normal distribution. Particularly, this is true for a logit transformation (commonly referred to as M-values in the context of methylation), which was advocated as more appropriate than beta values in an early work by Du et al. (2010). However, in the same work by Du et al., the authors showed that high heteroscedasticity, which violates the normality assumption, predominantly exists near the boundaries of the beta values (i.e. 0 and 1). That is, methylation sites that tend to be consistently methylated (i.e. mean level near 1 across the samples in the data) or consistently unmethylated (i.e. mean level near 0) are those who violate normality, whereas the rest of the sites demonstrate an approximately linear relation with their respective M-values, which eventually leads to the same performance in statistical testing (Du et al. 2010). This result can be explained by the relatively low variance observed in a given typical methylation site across individuals (albeit bounded to the range between 0 and 1).

The presumption that consistently methylated and consistently unmethylated sites are less likely to harbor biological signals of interest has led many investigators to exclude these sites from their analysis (e.g., Fernandez et al. 2012, Zou et al. 2014, Rhead et al. 2017, Lutsik et al. 2017, and others). Similarly, we exclude such sites from our analysis (see "Data sets" under Methods), which therefore alleviates the potential severe violation of the normal approximation.

Crucially, while a standard differential methylation analysis may be indifferent to the interpretation of the methylation values under test (in which case we may opt to use M-values over the more intuitive beta values), modeling bulk data as coming from a mixture of sources (cell types) requires interpretability, which is not given by M-values. Specifically, modelling the bulk mixture as a weighted combination of cell types would no longer have a natural mathematical interpretation under a logit transformation. For that reason, all previous works for tackling tissue heterogeneity in methylation have considered beta values rather than M-values (e.g., Houseman et al. 2012, 2014, 2016, Zheng et al. 2014, Rahmani et al., 2016, 2018, Waite et al. 2016, Lutsik et al. 2017, Teschendorff et al. 2017, and others).

In order to better reflect the above to the readers, we revised the paragraph discussing the appropriateness of the normality distribution in our context (see “The model” under Methods). It now reads:

“Admittedly, normality may not hold for values near the boundaries (i.e. sites with mean methylation levels approaching 0 or 1); this can be addressed by applying variance stabilizing transformations such as a logit transformation (commonly referred to as M-values in the context of methylation) [Du et al. 2010]. However, in practice, we ignore such consistently methylated or consistently unmethylated sites, which results in a set of sites that demonstrate an approximately linear relation with their respective M-values [Du et al. 2010]. This makes the normality assumption reasonable and therefore widely accepted in the context of statistical analysis of DNA methylation.

- The authors use Reinius data for the basis of simulation, and mention age and sex as related with DNA methylation. Reinius cell-type-specific data are from exclusively male subjects.

We agree with the reviewer that the Reinius et al. data is limited, as it does not reflect all the variability that we expect to have within a population. Yet, this valuable data provides a unique source for comparing cell-type-specific methylomes of several individuals across the main cell types in whole-blood. In order to address this genuine limitation of the data, we considered the case where methylation-affecting factors (such as age and sex) affect methylation levels. In all of our previous experiments, as well as the additional experiments we describe here, we use the Reinius et al. data for learning a general (i.e. across all individuals) background distribution, which is then assumed to be affected per individual by the individual-specific characteristics; these emulate the effect of methylation-altering sources of variation such as age and sex (see “Data simulations” under Methods).

As part of our simulation study, we show that TCA can in fact learn the effect of methylation-altering factors (Supplementary Figures S3 and S4). In a new set of additional experiments, we now also present a re-analysis of the power analysis while adding such methylation-altering factors. These experiments show that TCA remains effective even in the presence of such factors (Supplementary Figure S7).

- **There are some limitations of using the three dimensional tensor in the application setting of peripheral blood where there are approximately six major components. There is a use of k=6 in Figure 3, but additional detail regarding the noted limitation in the discussion (line 250) regarding “lowly abundant” cell types would add value.**

In the terminology of the TCA framework, we consider a *three-dimensional tensor* to represent the (unobserved) full data which is assumed to be consisted of individuals X sites X cell types information (i.e. three dimensions). The parameter k, which determines the number of assumed cell types, is set in our experiments to either 3 or 6 (each experiment is performed twice; under k=3 and under k=6). We agree with the reviewer that the assumption of k=6 better represents real whole-blood data (upon which we rely in our simulations). However, we consider the k=3 scenario, in addition to the more natural k=6 scenario, in order to demonstrate the performance of TCA in the case where the assumption of k=3 is appropriate. Put differently, the experiments with k=3 merely represent the case where TCA is applied to a tissue which is less heterogeneous than whole-blood. The use of the rich whole-blood reference for the data simulation in these experiments is therefore meant only for the purpose of choosing parameters that are realistic as much as possible.

Importantly, we have substantially revised the first part of the Results section in order to improve clarity, and we believe that now it better reflects the components of the TCA model and its relation to previous approaches. Particularly, we provide a new figure that illustrates the conceptual difference between TCA and previous methods (Figure 2).

The limitation of TCA with lowly abundant cell types that we noted in the discussion refers to a conclusion we observed in an analysis that was stratified by cell type, wherein the power of TCA under the scenario of a single cell type with a non-zero effect size is shown to increase for highly abundant cell types compared with lowly abundant cell types (Supplementary Figure S9; we edited the figure to include mean abundance for each cell type). This result is expected, given that lowly abundant cell types contribute weaker signals in bulk data compared with highly abundant cell types. We now further clarify this in the text (see “Applying TCA for detecting cell-type-specific associations in epigenetic studies” under Results):

“Finally, we performed an additional power analysis stratified by cell types, which, once again, showed that TCA robustly outperforms the alternative standard regression approach (Supplementary Figures S9 and S10). As expected, this analysis further revealed that under the scenario of a single cell type with a non-zero effect size, TCA achieved better power when the effect sizes were in highly abundant cell types (as opposed to lowly abundant cell types).

- **Why would the authors assume that each cell type has the same effect size if there are known differences in abundance and methylation is usually present or absent?**

The experiment with the assumption of the same effect size in all cell types was originally mainly designed for contrasting TCA with a commonly applied regression analysis under a scenario which is intuitively favorable for regression, which does not model cell-type-specific signals. In other words, this experiment mainly serves as a mean for emphasizing to the readers the need for a richer model.

Clearly, a model where all cell types have the same effect size is just one possible model. We therefore also considered and extensively evaluated the more interesting general case of different effects in different cell types (see Figure 3, and Supplementary Figures S8, S9, and S10, and additional experiments we now added and summarized in Supplementary Figure S7). We have substantially revised the first part of the Results section and the paragraph describing the power analysis (see "Applying TCA for detecting cell-type-specific associations in epigenetic studies" under Results) in order to better reflect all of the above.

Lastly, we would like to note that differences in abundance of the cell types do not necessarily imply different effect sizes for these cell types. There is no apparent reason to assume that the methylation levels of a particular cell type are, in general (i.e. globally in the genome), related to the abundance of that cell type. In fact, considering a model in which the abundance of a cell type is related to the methylation level of the cell type is bound to fail using a standard regression approach, due to an expected massive inflation in type 1 error in this case. For more on this and a demonstration using real data, please see our previous experiments described under "Cell-type-specific differential methylation analysis in immune activity" (Results), which we now revised for improved clarity.

- **It appears that the TCA approach testing cell-specific methylation does not inflate multiple comparisons. Please comment and add to strengths as appropriate**

As the reviewer correctly indicated, TCA controls adequately for false positive rate. We now reflect this in the Discussion:

"Importantly, we found that TCA is substantially superior to a standard regression analysis of bulk data, while adequately controlling for false positive rate, even in the case where all cell types share the same effect size."

- **Characterizing cell proportion estimates in multiple places as potentially noisy or "just moderately reasonable" is unclear, does it mean to suggest limited accuracy of the standard Reinius library applied using the Houseman approach?**

We thank the reviewer for pointing out this confusing point in the manuscript. Any computational method for estimating cell-type composition has a limited accuracy. This is true for both reference-based methods (which may be limited, for example, due to a low-quality reference or

in cases where the samples in the study are not well-represented by the reference) and reference-free methods (which, for example, may capture a convolution of the cell-type proportions with other factors of variation in the data). We therefore considered in our experiments the case where only noisy estimates of the cell-type proportions are available (under a wide range of noise level), and we show that TCA performs well even in those cases (for example, see Figure 3). In order to provide a direct evidence (i.e. beyond the power simulations) that TCA in fact allows to improve noisy cell-type proportion estimates that are given as an input, we now further demonstrate the ability of TCA to improve these estimates (Supplementary Figure S2).

In order to clarify the above, we revised the relevant paragraph in the Results section, which now reads as follows:

“Importantly, TCA requires knowledge of the cell-type proportions of the individuals in the data. These can be computationally estimated using either a reference-based supervised approach [Houseman et al. 2012] or a reference-free semi-supervised approach [Rahmani et al. 2018]; current reference-free unsupervised methods, however, are unable to provide reasonable estimates of cell-type proportions but rather only linear combinations of them [Rahmani et al. 2018]. Notably, in cases where only noisy estimates of the cell-type proportions are available (i.e. owing to inaccuracies of the computational method used for estimation), they can be used for initializing the optimization procedure of the TCA model, which can then provide improved estimates (Supplementary Figure S2). As a result, as we show next, TCA performs well even in cases where only noisy estimates of the cell-type proportions are available.”

In addition, we now avoid the previous confusing terminology (“moderately reasonable”). Particularly, we rewrote the relevant sentence in the Discussion, which now reads as follows:

“As we showed, this allows TCA to provide good results even when just noisy estimates of the cell-type proportions are available.”

- For the power simulations I am troubled by the replacement of each sample with randomly sampled proportions, this may result in overestimates of power as the true biological distribution would not be random within the Dirichlet.

TCA does not assume that the cell-type proportions are coming from a Dirichlet distribution but rather treats them as fixed values, and therefore using Dirichlet for sampling proportions is not expected to affect the performance of TCA. Notably, sampling from a Dirichlet was mainly done for convenience and for the fact that Dirichlet parameters fitted for a large sample of real blood cell counts (n=595) are available (Rahmani et al. 2018). That said, we agree with the reviewer that considering a different distribution for the proportions is of interest. We therefore now include additional experiments where we take a nonparametric approach and sample proportions from cell-type proportions that were estimated from real data using a reference-based approach (see “Applying TCA for detecting cell-type-specific associations in epigenetic studies” under Results):

“Repeating these experiments while including cell-type-specific affecting covariates and under a non-parametric distribution of the cell-type proportions (i.e. rather than a parametric one) demonstrated similar results (Supplementary Figure S7).”

For the full details regarding these experiments please see “Data simulations” and “Power simulations” under Methods.

- Similarly, the methods indicate better estimates in “more abundant” cell types (308-309), can the authors be more specific?

This general point was made following the result in equation (10) (Methods), showing that larger values of w_{hi} (which would correspond to high abundance of cell type h in individual i in the context of methylation studies) would result in a narrower distribution (i.e. lower variance), which implies lower uncertainty. In such cases, we therefore expect TCA to perform better at extracting cell-type-specific signals. We further refer the readers to the Supplementary Note, where we carefully construct equation (10) and discuss this issue in more details (see “Extracting underlying signals from convolved signals using TCA” under Supplementary Note).

- The details of what the TCA modeling approach provides as outputs are not entirely clear.

We thank the reviewer for raising this important issue. We now include a new illustration of the TCA model and highlight its distinction from a traditional decomposition approach (Figure 2). We moved the previous figure to the Supplementary Figures as an additional, more detailed view of the TCA model (Supplementary Figure S1). We further substantially revised the first part of the Results section for better clarity.

- Language “Standard approach” is often used and in certain places not clear, specify where possible (e.g. line 125, 230-231)

We replaced “standard approach” with “standard regression approach” where appropriate.

- The application of the approach is interesting, specifying the modeling approaches used in figure 4 panels would be helpful. An application to data with

We changed the labels of Figure 4 to improve clarity.

- **Another noted limitation in the discussion is covariance of cell type abundance, given the Dirichlet distribution this will often be the case and prior knowledge of how specific cell types co-vary could in the future help address the limitation and could be noted.**

We thank the reviewer for raising this important point. We revised to Discussion to reflect that (fourth paragraph). It now reads:

“Our experiments and mathematical results show that TCA can extract cell-type-specific signals from abundant cell types better compared with lowly abundant cell types. Another potential limitation is expected to be in the case where the proportion of one cell type strongly covary with the proportion of a second cell type. In case of a true association in just one of the two cell types, performing a marginal association test on each cell type separately might fail to effectively distinguish between the signals of the two cell types and report an association in both cell types. In light of these limitations, we suggest that future studies include small replication data sets from sorted or single cells. Future work might be able to alleviate this issue by modeling the covariance of the cell-type proportions.”

- **Additional opportunities for implementation of the method (R) would be most welcomed**

We will make an additional implementation of TCA in R available upon acceptance of the manuscript.

Reviewer #2 (Remarks to the Author):

In this manuscript, Rahmani et al. proposed an epigenome-wide association study in the cell-type-specific resolution using a so-called Tensor Composition Analysis (TCA) approach. The major contribution of their work is the using of bulk data instead of cell sorting or single-cell data, which are currently infeasible in large-scale study. The method they used, TCA, is a natural generalization (3D version) of the typical Nonnegative Matrix Factorization, which is widely explored in many epigenetic research fields (Houseman et al., 2012, 2016; Lutsik et al, 2017; Onuchic et al., 2016; Rahmani et al, 2018). I have the following major concerns, mainly in the methodology part.

- 1. The authors made a normality assumption on the methylation level of a given CpG sites within a particular type of cells. It is not exactly true to the best of my knowledge. Methylation levels vary between 0 and 1, and thus are more suitable to be modeled as the**

Beta distribution. If assuming the normal distribution, the authors should do some sort of data transformation on the raw beta values first, e.g., logit or arcsine.

We thank the reviewer for pointing out this important issue. Please see our elaborated response to the first comment made by Reviewer 1 regarding the normality assumption in TCA.

2. TCA require the cell-type proportion information as the input. However, in most cases, this kind of information is not available. There are two types of methods for estimation cell-type proportions from DNA methylation data. One is called 'reference-based', which is accurate but needs reference profiles of the constitutional components, and thus only available for blood samples or infiltrating immune cells. The other does not require the reference data but the estimation accuracy is quite low. So under such circumstance, the method proposed in this manuscript only works for blood samples.

We would first like to stress out that even if TCA could only be applied to blood, it still provides a major advance, since TCA allows to perform a deconvolution of bulk methylation into cell-type-specific signals, while previous methods only estimate the cell-type composition (see Figure 2 in the revised manuscript). Since currently methylation data are most commonly probed from blood, and since blood is the most widely used tissue in the clinic, we expect that TCA will greatly benefit most of the methylation studies.

As the reviewer rightfully pointed out, reference data is currently primarily available for blood and not for other tissues, thus partially limiting - yet not eliminating (e.g., Rahmani et al., 2018) - our ability to provide accurate estimates for tissues other than blood. Importantly, we address these cases where only noisy estimates of the cell-type proportions are available. Particularly, TCA allows to re-estimate the cell-type proportions that are given as an input to the algorithm. As we show in our analyses, this allows TCA to perform almost as good as in the case where the real cell-type proportions are known (Figure 3 and Supplementary Figures S7 and S9). We now further add an additional experiment that directly shows that TCA improves noisy estimates of the cell-type proportions (Supplementary Figure S2).

In order to clarify the above, we revised the relevant paragraph in the Results section, which now reads as follows:

“Importantly, TCA requires knowledge of the cell-type proportions of the individuals in the data. These can be computationally estimated using either a reference-based supervised approach [Houseman et al. 2012] or a reference-free semi-supervised approach [Rahmani et al. 2018]; current reference-free unsupervised methods, however, are unable to provide reasonable estimates of cell-type proportions but rather only linear combinations of them [Rahmani et al. 2018]. Notably, in cases where only noisy estimates of the cell-type proportions are available (i.e. owing to inaccuracies of the computational method used for estimation), they can be used for initializing the optimization procedure of the TCA model, which can then provide improved

estimates (Supplementary Figure S2). As a result, as we show next, TCA performs well even in cases where only noisy estimates of the cell-type proportions are available.”

Please also see the third paragraph of the Discussion section for further related discussion and considerations.

Finally, as more reference data and cell counts (which were shown to allow estimation of cell proportions without the need for reference data by applying a semi-supervised approach; see Rahmani et al. 2018) will become available, we expect that the TCA framework will allow to incorporate this information and further improve accuracy and power under more tissue types and conditions.

3. Along the second concern, if the estimation of cell-type proportion is not reliable, how about the accuracy of the EWAS result? The authors should show the robustness of their method under different sources and levels of noise.

Please see our response to the previous comment. Particularly, we evaluate the effect of inaccurate input of the cell-type proportion estimates (using a wide range of noise) on EWAS under different settings (Figure 3 and Supplementary Figure S9) and under several scenarios (different effect sizes for different cell types, a single effect size for all cell types, and a single effect size in a single cell type). These experiments show that TCA overcomes such inaccuracies and provides similar performance to the case where the real cell-type proportions are known. Please see “Power Simulations” under Methods for more details about these experiments and introducing noise into the cell-type proportions.

We now further add an additional experiment in order to directly show that TCA improves noisy estimates of the cell-type proportions (Supplementary Figure S2), and an additional experiment where the cell-type proportions are not drawn from a Dirichlet distribution but rather from cell-type proportions that were estimated from real data using a reference-based approach (Supplementary Figure S7).

4. In the real data application part, how did the authors determine the optimal value of k , i.e., the number of components in tissue mixture?

Throughout our experiments in the paper, the number of components (cell types) in the mixture is set according to the number of cell types for which proportions are given to the TCA algorithm as an input. Specifically, in the analysis of real data, we considered the six main blood cell types for which we obtained estimates using a reference-based approach (see “Implementation of TCA” under Methods).

We have substantially revised the first part of the Results section in order to improve clarity, and we believe that now it better reflects the components of the TCA model and its relation to previous approaches. Particularly, we provide a new figure that illustrates the conceptual difference between TCA and previous methods (Figure 2).

5. The authors should specify the computational complexity of their method and compare with other EWAS methods.

TCA involves a non-convex optimization, which requires numerical methods for non-linear optimization (see "Inferring the parameters of the model" and "Marginal test for the effect size of a particular cell type" under Supplementary Note). This greatly complicates any theoretical analysis of the computational complexity of the method. Importantly, we note that none of the existing methods provides output that is equivalent to that of TCA (see Figure 2 in the revised manuscript for clarification), and therefore we believe that it would be impossible to interpret the results of any comparison with existing methods. To improve clarity, we now provide a new Figure 2 that we believe better illustrates the conceptual difference between TCA and traditional decomposition methods.

Minor point:

- 1. Page 8: Index of Supplementary figure S4 is missing.**
- 2. Page 9: Figure 2 is vague. A carefully designed and more informative workflow of the method is needed.**

We thank the reviewer for these comments. We corrected the missing index, and we now include a new illustration of the TCA model and highlight its distinction from the traditional decomposition approach (Figure 2). We moved the previous figure to the Supplementary Figures as an additional, more detailed view of the TCA model (Supplementary Figure S1).

Reviewer #3 (Remarks to the Author):

The manuscript entitled “Cell-type-specific resolution epigenetics without the need for cell sorting or single-cell biology” by Elior Rahmani and colleagues proposed Tensor Composition Analysis (TCA), a new mathematical approach, to identify cell-type specific methylation changes from bulk epigenetic data without cell sorting. Cellular heterogeneity is a major problem impeding the application and interpretation of epigenome-wide association studies (EWAS). It would be of great importance if cell-type specific signatures could be extracted and explored from bulk data. While the topic is very important and the idea is interesting, there are several major issues with the manuscript.

Major issues:

1. Benchmarking and a more detailed data analysis are needed. For instance, the authors indicated poor performance of TCA for lowly abundant cell types and for the situation of strongly correlated cell types in the discussion. However, detailed evaluation on these situations using simulated data were not provided.

Please see our response to the third comment made by Reviewer 1 regarding the case of lowly abundant cell types. Particularly, we consider an analysis stratified by cell types to show that the power of TCA under the scenario of a single cell type with a non-zero effect size is shown to increase for highly abundant cell types compared with lowly abundant cell types (Supplementary Figure S9). This result is expected, given that lowly abundant cell types contribute weaker signals in bulk data compared with highly abundant cell types. Importantly, although the difference in power is notable, TCA still allows a dramatic improvement over a standard regression approach.

We revised the text in order to clarify the above (see “Applying TCA for detecting cell-type-specific associations in epigenetic studies” under Results):

“Finally, we performed an additional power analysis stratified by cell types, which, once again, showed that TCA robustly outperforms the alternative standard regression approach (Supplementary Figures S9 and S10). As expected, this analysis further revealed that under the scenario of a single cell type with a non-zero effect size, TCA achieved better power when the effect sizes were in highly abundant cell types (as opposed to lowly abundant cell types).”

We would like to clarify that TCA does not demonstrate a poor performance in the case where cell types covary in their proportions. Particularly, in all of our previous simulations, the cell-type proportions we used had a non-trivial covariance structure that was induced by the Dirichlet distribution (which we set according to parameters that were estimated from a large sample of real blood cell counts; Rahmani et al. 2018; see “data simulations” under Methods). We now further include experiments with cell-type proportions based on reference-based estimates from real data (i.e. instead of generating cell proportions from a Dirichlet distribution). These experiments show that TCA performs well when considering an empirical nonparametric distribution of the cell-type proportions rather than a parametric one (Supplementary Figure S7; for more details see “data simulations” under Methods).

Our indication of the possible limitation in the case of a strong correlation between cell types was made in order to acknowledge the possible problem of cell types identifiability. Particularly, as correctly recognized by the reviewer in comment 5, in the case where two cell types are highly correlated in their proportions, confidently attributing an association to a particular cell type may not be possible under the current framework, a point that we explicitly acknowledge in the Discussion. We further believe that this potential limitation, as well as the limitation arising from the cell type abundance, form a good direction for future work that may improve upon the performance of TCA and address its current limitations. We revised the text in the Discussion section in order to better reflect and clarify that. It now reads:

“Our experiments and mathematical results show that TCA can extract cell-type-specific signals from abundant cell types better compared with lowly abundant cell types. Another potential limitation is expected to be in the case where the proportion of one cell type strongly covary with the proportion of a second cell type. In case of a true association in just one of the two cell types, performing a marginal association test on each cell type separately might fail to effectively distinguish between the signals of the two cell types and report an association in both cell types. In light of these limitations, we suggest that future studies include small replication data sets from sorted or single cells. Future work might be able to alleviate this issue by modeling the covariance of the cell-type proportions.”

Finally, we would like to note that for the purpose of a clear and simple presentation of the Results section, we included in the Methods many of the details of each of the analyses we performed. We believe that this is vital for the clarity of this paper, owing to the unintuitive concept we propose with TCA. For a full description of the power study please see the paragraphs "Data simulation" and "Power simulations" and for a full description of the real data analysis please see the paragraphs "Analysis of immune activity" and "Analysis of rheumatoid arthritis" under the Methods section.

2. In the past several years, numerous mathematical approaches have been proposed for cell type deconvolution and subsequent ewas analyses, including both reference-based (eg. CIBERSOFT, RPC) and reference-free (eg. SVA, RefFreeEWAS, ISVA, EWASher, ReFACTor, RefFreeCellMix) approaches. They all performed better compared to a simple linear regression model. Even though these approaches assume the same effect size for all cell types, it is critical for the authors to evaluate the performance of TCA in the context of these for Scenario II in Figure 3.

We thank the reviewer for raising this point, which we believe is due to lack of clarity in the previous version of the manuscript. All previous methods for estimating cell-type composition from methylation were designed for directly learning cell-type proportions or finding appropriate surrogates that can allow to control for type 1 error in association studies, which was shown to be a major issue in cases where the phenotype of interest is correlated with the cell-type composition (Jaffe and Irizarry, 2014). TCA, on the other hand, suggests a new concept, in which we consider the cell-type-specific signals. We now reflect these differences more clearly in Figure 2, where we show the conceptual differences between TCA and previous methods. As part of our previous set of experiments we demonstrated this conceptual difference between TCA and a standard analysis by considering immune response as a phenotype. Since the phenotype is defined by the cell composition in this case, applying any standard method (either reference-based such as CIBERSORT or reference-free such as SVA) for correction would practically eliminate all the true signal (which is exactly what a standard method would be removing). In order to further improve clarity, we revised the subsection describing this experiment (see “Cell-type-specific differential methylation analysis in immune activity“ under Results).

Our original intent in the experiment with the same effect size for all cell types was to deliver the message that a scenario in which all cell types have the same effect is intuitively expected to be a favorable scenario for a standard regression analysis, which does not model cell-type-specific signals. Importantly, this scenario was mainly designed to serve as a mean for emphasizing to the readers the need for richer models for heterogeneous methylation data (please also see our response to the fourth comment made by Reviewer 1), and we consider other scenarios as well (different effects in different cell types and a single effect in a single cell type).

We would like to acknowledge and clarify that the previous version of the manuscript could have been misinterpreted with regard to the motivation for the experiments with the same effect size in all cell types, potentially leading the reader to think that all previous methods assume a single effect in all cell types. Previous methods did not model cell-type-specific signals, and hence these methods also did not assume a single effect size (but rather a single effect size with a “tissue-level” methylation). We revised the text in order to avoid this confusion (see "Applying TCA for detecting cell-type-specific associations in epigenetic studies" under Results):

“Remarkably, TCA demonstrated the highest improvement in a scenario where all cell types had the exact same effect size, although this is intuitively a favorable scenario for a standard regression analysis, which does not model cell-type-specific signals (Figure 3)”.

3. It is unclear to me how the TCA works based on Figure 2 and its legend. Even though it is described in details in the supplementary note, it is important for readers to understand the main idea of TCA from the main text and the figure.

We thank the reviewer for raising this important issue. We now include a new illustration of the TCA model and highlight its distinction from a traditional decomposition approach (Figure 2). We moved the previous figure to the Supplementary Figures as an additional, more detailed view of the TCA model (Supplementary Figure S1).

4. The author claimed in the abstract that “we show mathematically and experimentally that cell-type-specific methylation levels of an individual can be learned from its tissue-level bulk data, as if the sample has been profiled with a single-cell resolution and then signals were aggregated in each cell population separately”. This is a gross overstatement. First, no experiments were performed in the manuscript. Second, the results generated from TCA model were not compared to results from single cell data. In fact, the TCA model makes the normality assumption for methylation, while methylation in a single cell follows binary distribution. Thus, the TCA model, theoretically, could not be extended to single cell.

We thank the reviewer for raising these concerns, which we believe point out a confusion due to lack of clarity in the text. The sentence referred by the reviewer was not meant to be taken literally, but rather was designed for delivering the (fairly complicated) key idea of the paper using an abstraction of the methodology; based on feedback from several of our colleagues, we believe that this kind of abstraction is crucial for an exposition that will be clear to many of the readers. To accommodate the risk of misinterpretation, we revised the above sentence to reflect the fact that the results we present in the paper are empirical (rather than “experimental”, which might be perceived differently than our intent) and to clarify that we refer to single-cell experiments merely as an analogy and for the sake of explanation rather than for making any technical claim:

“Here, we show mathematically and empirically that cell-type-specific methylation levels of an individual can be learned from its tissue-level bulk data, conceptually emulating the case where the individual has been profiled with a single-cell resolution and then signals were aggregated in each cell population separately.”

In addition, we have substantially revised the first part of the Results section in order to improve clarity, and we believe that now it better reflects the idea behind TCA and the components of the model.

5. In the reference-free situation, it is unclear to me how TCA extracted cell-type specific methylation signature was assigned to one particular cell type when there is another cell type with a similar cell proportion. This is particularly worrisome when looking at cell-type-specific analysis using TCA in Figure 4f. Several significant CpGs identified from this approach, but failed to be replicated in the claimed cell type using sorted data, were actually significant in other cell types (eg. cg11767757 in CD14+ and CD19+ cells; cg16411857 in CD4+ cells). To me, this cannot be explained by the small sample size as claimed by the authors.

6. The most unique feature of TCA, as claimed by the authors, is the ability to identify cell-type specific methylation changes. However, based on the results from Figure 4d and 4f, cell-type specific approach did not identify any new significant hits outside of the HLA region. Combining with the fact of high false positive rate of assigned cell types by TCA as discussed in point 5, this advantage of TCA model is debatable.

We thank the reviewer for raising these concerns, which we believe are in part due to lack of clarity in the previous version of the manuscript.

In the previous version of the manuscript, we considered a Bonferroni-adjusted threshold for reporting replication rates using the sorted data. However, owing to the limited sample size of the sorted data, taking this approach does not reveal the full spectrum of consistency between the associations reported by TCA and the corresponding results of these CpGs in the sorted data. In order to demonstrate this better, we now consider a more permissive threshold for the evaluation of the CpGs detected by TCA. Specifically, we quantify the number of CpGs that

were verified to demonstrate a significant p-value at level 0.05 (i.e. not adjusted for multiple comparison).

Given this new perspective (which is not expected nor claimed to satisfy the typical family-wise error rate), it is clear that the CpGs reported by TCA are enriched for low p-values. Particularly, we consider two models using TCA: a joint analysis (that allows to contrast TCA with a standard regression analysis) and a cell-type-specific (marginal) analysis for each one of the three cell types for which we have sorted data (the sorted data by Rhead et al.). We found that 11 of the 15 CpGs reported by the joint analysis of TCA had a significant p-value at level 0.05 in at least one of the three cell types in the Rhead et al. data. In the marginal analysis, we found that 4 of the 6 associations in CD4+ and 4 of the 8 associations in CD14+ had a significant p-value at level 0.05, with all associations (including the single association in CD19+) having overall low p-values (p-value \leq 0.35 for all 15 associations; see Supplementary File 2). Following the enrichment in small p-values, we further considered a false discovery rate (FDR) criterion for the entire set of 15 associations, which revealed significant q-values at level 0.05 for all 15 associations. An additional dataset with sorted CD4+ methylation from an RA study by Guo et al. (n=24) was found to be consistent (p-value $<$ 0.05) with 3 of the 4 CD4+ associations that were detected in the Rhead et al. data (see Supplementary File 2). Our results further show that among the associations that are consistent with the sorted data, a joint analysis using TCA detected associations in 5 loci outside the HLA region that were not reported by a regression analysis, and the marginal analysis using TCA detected an association in one locus outside the HLA region that was not reported by a regression analysis.

We believe that this new presentation of the results provides a better perspective that reflects a high consistency between TCA and the ground truth that is defined by the available sorted cells data, thus demonstrating that the associations detected by TCA are meaningful. Importantly, since a family-wise error rate is no longer satisfied, we do not consider these results as replicated results but rather as further (significant) statistical evidence that is consistent with TCA.

As indicated by the reviewer, a notable result of our analysis shows that in some cases TCA seems to fail in attributing the correct cell types to associations. Specifically, we found that two associations (cg16411857 and cg22812614) were attributed by TCA to CD4+, however were supported by the sorted data to be CD14+ specific, and another association (cg11767757) was attributed to all cell types, however, was only supported by the sorted data to be CD14+ specific. Failing to tag the correct cell types that are driving an association can be explained by the cell type identifiability problem that we acknowledge in the Discussion and that was recognized by the reviewer. As we explain in our response to the first comment made by the reviewer, this is a limitation of the current model; we suggest this as a possible direction for future work (see the fourth paragraph in the Discussion section). Importantly, we note that this limitation, together with the fact that each of the datasets (i.e. the whole-blood and the two sorted datasets) was collected from a different population, suggest possible explanations for the several associations that were found by TCA but not supported by the available sorted data. We

added the following paragraph in order to reflect the above (see “Cell-type-specific differential methylation analysis in rheumatoid arthritis” under Results)

“Finally, we note that the lack of evidence for some of the associated CpGs may be explained in part by the fact that each data set was collected from a different population; specifically, Liu et al. studied a Swedish population, Rhead et al. studied a heterogeneous European population, and Guo et al. studied a Han Chinese population. Another possibility is that TCA did not attribute the correct cell types to some of the associations. A support for this possibility is given by the fact that two associations (cg16411857 and cg22812614) were attributed to CD4+, however were supported by the sorted data to be CD14+ specific, and another association (cg11767757) was attributed to all cell types, however, was only supported by the sorted data to be CD14+ specific.”

Altogether, our results show that the sorted data is overall consistent with the results reported by TCA, therefore demonstrating a compelling evidence for the utility of TCA in providing insights with cell-type-specific resolution. While the current framework has its limitations, we believe that it can greatly benefit many studies with heterogeneous methylation. Importantly, as we note in the Discussion, future work may allow to address the limitations of the current framework and to further improve upon its performance.

We have substantially revised the subsection describing the RA analysis in the paper in order to allow a better and clearer presentation of the results and all of the above points.

7. The analysis in the last paragraph of the main text (page 16), analyzing the associations between methylation and expression, is quite problematic. The three CpG sites investigated are all located in the HLA region. There are many SNPs in the probes for these CpGs, which will greatly affect the methylation measurements. Similarly, the expression data used for analysis was generated from microarrays and the high genetic polymorphism within the HLA region will also impact the evaluation of gene expression. Thus, genotype, as a confounder, will lead to observed associations between methylation and gene expression.

We thank the reviewer for this thoughtful comment. This analysis aimed at providing more insights into the potential role of methylation in rheumatoid arthritis, given the known regulation interplay between methylation and expression (e.g., Moore, Thuc and Guoping, 2013). Importantly, as part of the data preprocessing steps, we excluded cross-reactive sites and polymorphic sites (i.e. those with a SNP in the target position), as was previously suggested by Chen et al. (2013), thus alleviating the risk that methylation measurements were artificially affected by genotypes. The reviewer suggests that genotypes can act as a confounder - in this case, since genotypes are causal with respect to the disease status, we argue that a correlation between methylation and expression is interesting, even if this correlation is entirely explained by genetics, as such a correlation may provide insights into the components and mediators in the causality network of the disease. That said, we believe that this analysis is not critical for evaluating the validity and performance of the methodology we propose, and given these

comments by the reviewer, we acknowledge the potential problems with the interpretation of results in this experiment and we therefore decided to omit this analysis from the paper.

References

- Du, Pan, et al. "Comparison of Beta-value and M-value methods for quantifying methylation levels by microarray analysis." *BMC bioinformatics* 11.1 (2010): 587.
- Fernandez, Agustin F., et al. "A DNA methylation fingerprint of 1628 human samples." *Genome research* 22.2 (2012): 407-419.
- Rhead, Brooke, et al. "Rheumatoid arthritis naive T cells share hypermethylation sites with synoviocytes." *Arthritis & Rheumatology* 69.3 (2017): 550-559.
- Zou, James, et al. "Epigenome-wide association studies without the need for cell-type composition." *Nature methods* 11.3 (2014): 309.
- Lutsik, Pavlo, et al. "MeDeCom: discovery and quantification of latent components of heterogeneous methylomes." *Genome biology* 18.1 (2017): 55.
- Houseman, Eugene Andres, et al. "DNA methylation arrays as surrogate measures of cell mixture distribution." *BMC bioinformatics* 13.1 (2012): 86.
- Houseman, Eugene Andres, John Molitor, and Carmen J. Marsit. "Reference-free cell mixture adjustments in analysis of DNA methylation data." *Bioinformatics* 30.10 (2014): 1431-1439.
- Houseman, E. Andres, et al. "Reference-free deconvolution of DNA methylation data and mediation by cell composition effects." *BMC bioinformatics* 17.1 (2016): 259.
- Zheng, Xiaoqi, et al. "MethylPurify: tumor purity deconvolution and differential methylation detection from single tumor DNA methylomes." *Genome biology* 15.7 (2014): 419.
- Rahmani, Elio, et al. "Sparse PCA corrects for cell type heterogeneity in epigenome-wide association studies." *Nature methods* 13.5 (2016): 443.
- Rahmani, Elio, et al. "BayesCCE: a Bayesian framework for estimating cell-type composition from DNA methylation without the need for methylation reference." *Genome biology* 19.1 (2018): 141.
- Waite, Lindsay L., et al. "Estimation of cell-type composition including T and B cell subtypes for whole blood methylation microarray data." *Frontiers in genetics* 7 (2016): 23.
- Teschendorff, Andrew E., et al. "A comparison of reference-based algorithms for correcting cell-type heterogeneity in Epigenome-Wide Association Studies." *BMC bioinformatics* 18.1 (2017): 105.
- Jaffe, Andrew E., and Rafael A. Irizarry. "Accounting for cellular heterogeneity is critical in epigenome-wide association studies." *Genome biology* 15.2 (2014): R31.
- Moore, Lisa D., Thuc Le, and Guoping Fan. "DNA methylation and its basic function." *Neuropsychopharmacology* 38.1 (2013): 23.
- Chen, Yi-an, et al. "Discovery of cross-reactive probes and polymorphic CpGs in the Illumina Infinium HumanMethylation450 microarray." *Epigenetics* 8.2 (2013): 203-209.

Reviewers' comments:

Reviewer #1 (Remarks to the Author):

The authors have been responsive to the comments in my review.

My remaining suggestions are that:

- lines 409-411 in the methods also be stated earlier in the methods section under The Model.
- Providing more detail of the power results for analyses stratified by cell type, (the language referring to Figures S9 and S10) would be helpful to readers, perhaps referring to specific cell types, their abundance levels, and power estimates

Reviewer #2 (Remarks to the Author):

The authors has addressed all my concerns.

I carefully evaluated the concerns the reviewer 3 and the authors' responses. I believe that the author did not satisfactorily address at least two questions.

1. In question 2, the reviewer asked the authors to compare their method with other existing methods in identifying cell-type-specific sites, which is a crucial step to justify a new proposed method. But the authors only compared it with naive model in Figure 3. By the way, I noticed two other methods, CellIDMC by Zheng et al. and HIRE by Luo et al. (see attached), that also work on the same problem. I think the author should further evaluate their method by comparing with these methods.

2. Question 6, the reviewer did not find new significant hits beside the HLA region. I am not an expert in this area, but I think the authors should have a more proper example to show their strength.

CellIDMC: <https://www.ncbi.nlm.nih.gov/pubmed/30504870>

HIRE: <https://www.biorxiv.org/content/biorxiv/early/2018/09/12/415109.full.pdf>

April 1st, 2019

Dear Editors and Reviewers,

We thank the reviewers for their comments. We have revised the manuscript to address the reviewers' questions and suggestions. Below we provide a point-by-point response to the reviewers' comments (our responses are in regular fonts). In addition, we include a manuscript file where the changes in the text are highlighted in a blue font.

Reviewer #1 (Remarks to the Author):

The authors have been responsive to the comments in my review.

My remaining suggestions are that:

- lines 409-411 in the methods also be stated earlier in the methods section under The Model.

Per the suggestion of the reviewer, we now also state the mean-based exclusion criterion (previously described only in lines 409-411) under the Model subsection:

"...in practice, we ignore such consistently methylated or consistently unmethylated sites (e.g., in our experiments we discarded sites with mean value higher than 0.9 or lower than 0.1), which results in a set of sites that demonstrate an approximately linear relation with their respective M-values."

- Providing more detail of the power results for analyses stratified by cell type, (the language referring to Figures S9 and S10) would be helpful to readers, perhaps referring to specific cell types, their abundance levels, and power estimates

We revised the text to include more details about this analysis. The revised paragraph now reads as follows (see final paragraph under the "Applying TCA for detecting cell-type-specific associations in epigenetic studies" subsection):

"Finally, we performed an additional power analysis stratified by cell types, which, once again, showed that TCA robustly outperforms the alternative approaches (Supplementary Figures S9 and S10). This analysis further revealed that under the scenario of a single causal cell type, TCA achieved better power when the causal cell type was highly abundant (as opposed to lowly abundant); these results are expected, given that bulk signals are mostly dominated by abundant cell types. For instance, considering a moderate effect size corresponding to a signal-to-noise ratio of 1, we found that TCA achieved a median power of 1 and 0.52 in granulocytes and CD4+ cells (the two most abundant cell types; mean abundance of 0.67 and 0.11, respectively), yet only a limited power in the less abundant cell types; for example, in the two least abundant cell types considered, B cells and NK cells (mean abundance 0.03 for both),

TCA could only achieve a median power of 0.08 and 0.03 under the same effect size (Supplementary Figure S9).“

Reviewer #2 (Remarks to the Author):

The authors has addressed all my concerns.

I carefully evaluated the concerns the reviewer 3 and the authors' responses. I believe that the author did not satisfactorily address at least two questions.

1. In question 2, the reviewer asked the authors to compare their method with other existing methods in identifying cell-type-specific sites, which is a crucial step to justify a new proposed method. But the authors only compared it with naive model in Figure 3. By the way, I noticed two other methods, CellDMC by Zheng et al. and HIRE by Luo et al. (see attached), that also work on the same problem. I think the author should further evaluate their method by comparing with these methods.

We thank the reviewer for this thoughtful comment. We now evaluate both CellDMC by Zheng et al. and HIRE by Luo et al. - both were not published at the time of the submission of our manuscript, but rather they were described in preprints - and we show compelling evidence that TCA substantially advances over these alternatives.

CellDMC was recently published (while our paper being under review), and therefore evaluating it as a possible alternative to TCA is appropriate. We now therefore include CellDMC in our benchmarking throughout the paper. We show that TCA substantially outperforms CellDMC under all the scenarios we considered in our simulation study, even though we provided CellDMC with the true cell-type proportions as an input (see revised Figure 3 and Supplementary Figures S7, S8, S9, S10, and the revised subsection “Applying TCA for detecting cell-type-specific associations in epigenetic studies” under Results). Notably, we evaluated TCA under different levels of noise injected into the cell-type proportions that were given as input, showing that TCA outperforms CellDMC even under the most noisy cases.

In the real data analysis, CellDMC was uncalibrated and failed to detect associations in the EWAS with immune activity (see revised Figure 4 and the “Cell-type-specific differential methylation analysis in immune activity” subsection under Results), and reported several associated CpGs in the rheumatoid arthritis analysis, none of which could be replicated using the sorted data (see revised Figure 4 and the “Cell-type-specific differential methylation analysis in rheumatoid arthritis” subsection under Results).

We further added the following paragraph to introduce the idea behind CellDMC to the readers (see the first subsection of the Results):

“Notably, in the context of differential gene expression analysis, it has been previously suggested that cell-type-specific effects can be estimated by treating a phenotype of interest as

a covariate (i.e. of the expression level) with potentially different effects on different cell types [Shen-Orr et al., Westra et al.]. Practically, this approach suggests to evaluate the effect of an interaction term (i.e. a multiplicative term) of the cell-type composition and the phenotype under a standard regression framework (i.e. by adding the interaction term to Equation (1) [Westra et al.]; equivalently, one may achieve the same goal by solving multiple decomposition problems (one for each possible value of the phenotype) [Shen-Orr et al.]. In fact, this concept was recently applied and reported in the context of DNA methylation in attempt to detect cell-type-specific differences in methylation [Zheng et al.]. However, as we demonstrate below, a more detailed model of the variation in bulk methylation data as described in this manuscript allows a substantial improvement in power.”

Similarly, we verified that TCA substantially outperforms HIRE (see Supplementary figures S13 and S14), however, given that it is not a standard practice to benchmark new methods with unpublished work, we did not include these results in the main analysis of the paper. We believe that a full benchmarking with unpublished work may result in an unfair evaluation of a method which may still be under development through a peer-review process. In fact, while we were able to apply the current version of the HIRE software to our relatively small simulated data sets, the software failed in operating on the large real data sets we analyzed. In order to balance between fairness towards a yet unpublished work and the need to show the uniqueness of the powerful TCA model, we refer to HIRE only in the Discussion section, where we refer to separate figures that demonstrate the differences in power between TCA and HIRE under our simulation study, while emphasizing that the results are based on the current version of their method:

“We further note that around the time of submitting this manuscript, another model similar to TCA appeared as a preprint by Luo et al. For completeness, we verified that TCA performs substantially better than the method by Luo et al. (Supplementary Figures S13 and S14; see Methods); given that the latter was not published by the time of submitting this manuscript, we separate this evaluation from the main benchmarking in our work.”

We further added an additional subsection under Methods, describing the application of CellDMC and HIRE in our study (see “Application of CellDMC and HIRE”).

Finally, the reviewer refers to the comment made by Reviewer 3 in the previous round of review regarding benchmarking with other existing methods. As far as we know, the only currently available methods that were designed for the detection of cell-type-specific associations with methylation are CellDMC and HIRE. We agree with the reviewer that other methods that model tissue heterogeneity in DNA methylation may potentially reveal cell-type-specific associations. However, all of these alternative methods, which were developed in the context of correction for cell-type composition, effectively look for surrogate components of the cell-type composition; these are then used in a standard analysis for adjusting for inter-individuals changes in cell-type composition. Thus, an upper bound for the performance of all of these methods in our case can be given by an analysis that perfectly adjusts for the true cell-type composition (i.e. rather than adjusting for noisy surrogates of the cell-type composition). In our experiments, we provided CellDMC with the true cell-type proportions as an input, and since CellDMC considers additive

effects of the cell type composition (i.e. it adjusts for cell-type composition, beyond considering interaction terms for detecting cell-type-specific associations), it allows CellDMC to achieve a perfect correction and therefore practically provides an upper bound for the performance of all other methods. We now clarify this point in the first part of the Result section:

“We next evaluated the performance of TCA in detecting cell-type-specific associations by simulating whole-blood methylation and corresponding phenotypes with cell-type-specific effects. We compared the performance of TCA with a standard regression analysis of the bulk levels and with the method CellDMC, an interaction-based test that was recently evaluated in the context of detecting cell-type-specific associations with methylation [Zheng et al.]. Notably, we provided CellDMC with the true underlying cell-type proportions as an input. Beyond introducing interaction terms into a standard regression framework, CellDMC also considers additive effects of the cell-type composition. Given the true cell-type proportions, it therefore achieves a perfect linear correction for cell-type composition. Hence, CellDMC practically reflects in our experiments an upper bound for the performance of any standard method that merely accounts for linear differences in cell-type composition across individuals.”

2. Question 6, the reviewer did not find new significant hits beside the HLA region. I am not an expert in this area, but I think the authors should have a more proper example to show their strength.

In fact, we did find a few associations outside the HLA region. We kindly refer the reviewer to the results presented in Figure 4 and to the associated results described in the subsection “Cell-type-specific differential methylation analysis in rheumatoid arthritis” under Results and in Supplementary File 2. Our results show that TCA with a joint analysis detected associations in 5 non-HLA loci that were not detected by a standard analysis and were consistent in the sorted methylation data (cg11767757, cg09716921, cg01823925, cg08913523, and cg09571369); TCA with a marginal analysis detected 4 non-HLA loci that were not detected by a standard analysis and were consistent in the sorted methylation data (cg11767757, cg08913523, cg05113410, and cg02131853).

We would like to further stress out that this analysis reflects a relatively unique opportunity to work on the same phenotype using both large whole-blood data (Liu et al., n=658) and a scarce, relatively large data set with methylation collected from several sorted leukocyte sub-types (Rhead et al., n=90). We believe that our multiple evidence from this analysis, combined with the analysis of the immune response (were we detected and replicated several associations including two novel replicating associations), and in conjunction with the extensive simulation study we performed, compose a set of compelling evidence that TCA substantially advances over the state-of-the-art in methylation association studies.

REVIEWERS' COMMENTS:

Reviewer #1 (Remarks to the Author):

The authors have addressed my comments

Reviewer #2 (Remarks to the Author):

The authors have satisfactorily addressed my concerns.